# Quantifying CH₄ emissions from coal mine aggregation areas in Shanxi, China using TROPOMI observations and the wind-assigned anomaly method

Qiansi Tu[1], Frank Hase[2], Kai Qin[3], Jason Blake Cohen[3], Farahnaz Khosrawi[4], Xinrui Zou[1], Matthias Schneider[1], Fan Lu[3]

[1]Tongji University, School of Mechanical Engineering, Shanghai, China
[2]Karlsruhe Institute of Technology (KIT), Institute of Meteorology and Climate Research (IMK-ASF), Karlsruhe, Germany
[3]China University of Mining and Technology, School of Environment and Spatial Informatics, Xuzhou, China
[4]Forschungszentrum Jülich GmbH, Jülich Supercomputing Centre (JSC), Jülich, Germany

*Correspondence to*: Qiansi Tu (tuqiansi@tongji.edu.cn), Kai Qin (qinkai@cumt.edu.cn)

**Abstract.**

China stands out as a major contributor to anthropogenic methane (CH₄) emissions, with coal mine methane (CMM) playing a crucial role. To control and reduce CH₄ emissions, China has made a dedicated commitment and formulated an ambitious mitigation plan. To verify the progress made, the consistent acquisition of independent CH₄ emission data is required. This paper aims to implement a wind-assigned anomaly method for the precise determination of regional-scale CMM emissions within the coal-rich Shanxi province. We use the TROPOspheric Monitoring Instrument (TROPOMI) CH₄ observations from May 2018 to May 2023, coupled with ERA5 wind and a bottom-up inventory dataset based on the IPCC Tier 2 approach covering the Changzhi, Jincheng and Yangquan regions of the Shanxi province. The derived emission strengths are $8.4 \times 10^{26}$ molec. s⁻¹ (0.706 Tg yr⁻¹, ± 25%), $1.4 \times 10^{27}$ molec. s⁻¹ (1.176 Tg yr⁻¹, ± 20%), and $4.9 \times 10^{26}$ molec. s⁻¹ (0.412 Tg yr⁻¹, ± 21%), respectively. Our results exhibit biases of -18%, 8%, and 14%, respectively, when compared to the IPCC Tier 2 bottom-up inventory. Larger discrepancies are found when comparing the estimates to the CAMS-GLOB-ANT and EDGARv7.0 inventories (64%-176%), suggesting that the two inventories may be overestimating CH₄ emissions from the studied coal mining regions. Our estimates provide a comprehensive characterization of the regions within the Shanxi province, contribute to the validation of emission inventories, and provide additional insights into CMM emissions mitigation.

## 1. Introduction

Methane (CH₄) is the second most important anthropogenic greenhouse gas (GHG) with a relatively shorter lifetime but a larger global warming potential than carbon dioxide (CO₂) (IPCC, 2014; Etminan et al., 2016). For this reason, efforts to reduce CH₄ emissions would be beneficial for rapid climate change mitigation in the short term. The atmospheric CH₄ is emitted from a variety of natural sources (accounting for 40%, e.g., wetlands, termites) and anthropogenic sources (accounting for 60%, e.g. industrial fossil fuel production and consumption, waste disposal, agriculture) (Saunois et al., 2020). Currently,

a significant fraction (~33% for the 2008-2017 decade) of global $CH_4$ emissions related to fossil fuels comes from the exploitation, transportation, and usage of coal (Saunois et al., 2020). China is one of the leading $CH_4$ emitters in the world and accounted for around 14-22% of global anthropogenic $CH_4$ emission (Janssens-Maenhout et al., 2019; Liu et al., 2021). China has demonstrated its commitment to addressing $CH_4$ emissions by signing key international agreements such as the Kyoto

Protocol in 1998 and the Paris Agreement in 2016, underscoring its commitment to global efforts in mitigating climate change. Additionally, in 2021, China committed to reduce $CH_4$ emissions under the Glasgow Agreement and intended to develop a comprehensive and ambitious National Action Plan with the goal of achieving a substantial impact on methane emission control and reductions in the 2020s (USDoS, 2021). Thus, the precise measurement of $CH_4$ emission changes is essential for determining the effectiveness of these commitments.

The anthropogenic $CH_4$ emissions in China increased by 40% in the 2000s (Liu et al., 2021), probably reflecting increasing coal production (Gao et al., 2021). Coal production in China reached 3.9 Gt in 2020, with approximately half of the coal being utilized for thermal power generation (National Bureau of Statistics of China, 2022). China's official GHG emission inventory (MEE, 2019) reports that the country's coal mine methane (CMM) emissions amounted to about 21 Tg in 2014, thus accounting for 38% of its total anthropogenic $CH_4$ emissions. China has submitted three versions of the National Communications on

Climate Change (NDRC, 2004, 2012; MEE, 2019b) and two reports of Biennial Update Reports on Climate Change since 2004 (NDRC, 2017; MEE 2019a), in which the estimated inventories of the CMM emissions are reported. The current CMM emission inventories are usually based on bottom-up data-based approaches, which involves identifying and quantifying the $CH_4$ emissions from each type of coal mine (Gao et al., 2020).

Mainland China's coal mines are spread across 26 provinces and were comprised of approximately 1000 coalfields and over

10,000 coal mines in 2011 (SACMS, 2012). The CMM emissions in China show unique characteristics and complexities, due to the large variability of the coal rank, capacity, geological conditions, and mining technologies of the numerous coal mines (Gao et al., 2020, 2021; Peng et al., 2016; Scarpelli et al., 2020). This large number of coal mines and the heterogeneity between them also induce considerable uncertainties in bottom-up estimates and are thus a challenge in achieving accurate CMM emissions estimates (Sheng et al., 2019). Qu et al. (2021) highlighted significant challenges in their satellite inversion

over southeast China characterized by elevated seasonal rice emissions that coincide with extensive cloud cover and potential misallocation of coal emission. A recent study from Chen et al. (2022) suggests a downward correction in CMM emissions (-15%) in China compared to the United Nations Framework Convention on Climate Change (UNFCCC) reports, partly driven by reductions in the Shanxi province. Zhang et al. (2021) documented an overestimation of anthropogenic emissions from China, revealing a 30% decrease in the posterior estimates, with approximately 60% of this downward correction attributed to coal

mining. Therefore, a strong demand exists for independent and objective verification of CMM emissions from local to regional scales based on atmospheric observations, which are commonly known as top-down approaches. The observations from satellites, e.g. the TROPOspheric Monitoring Instrument (TROPOMI) on board the Sentinel-5 Precursor satellite, provide due

to their global, high-resolution measurements, the ability to estimate the CMM emissions from regional scales (Sadavarte et al., 2021; Tu et al., 2022b; Chen et al., 2022) to a global scale (Shen et al., 2023).

This study conducts the wind-assigned anomaly method (Tu et al., 2022a, b) on TROPOMI $XCH_4$ observations derived from 2018 to 2023 for three subregions in the Shanxi province to determine the CMM emissions over that period. Shanxi province is known for its abundant coal reserves and is considered one of the coal-richest provinces in China. The coal production in Shanxi exceeded 1 billion tons in 2021, accounting for nearly one-third of the country's total coal output and 12% of the global output. This highlights the significant role of Shanxi province in China's energy sector and emphasizes the

importance of estimating CMM emissions from the mining activities in the region. In this work, the emission estimation method and the TROPOMI dataset are introduced in Sect. 2. In Sect. 3 we present the results of the TROPOMI observations and three different inventories used for comparison, followed by estimated CMM emissions over three subregions. An uncertainty analysis based on a dispersion model, wind information and inventory are also performed in this section. A conclusion is given in Sect. 4.

**2. Data and method**

**2.1 TROPOMI dataset**

    Launched in October 2017, the TROPOMI instrument is an imaging spectrometer which is designed to view the Earth in nadir direction. The instrument utilizes passive remote-sensing techniques to measure the backscattered solar radiation across the ultraviolet (UV), visible (VIS), near-infrared (NIR), and short-wave spectral (SWIR) bands (Veefkind et al., 2012). The

instrument is capable of providing an unprecedented combination of high spatial resolution ($5.5 \times 7$ km$^2$) and complete daily global coverage of the $CH_4$ total column-averaged dry-air mole fraction ($XCH_4$) (Veefkind et al., 2012; Lorente et al., 2021). The RemoTec algorithm, which has been widely utilized in deriving $CH_4$ and $CO_2$ from the Greenhouse Gases Observing Satellite (GOSAT) (Butz et al., 2011; Guerlet et al., 2013), is also deployed here to retrieve $XCH_4$ from TROPOMI measurements. These measurements capture sunlight backscattered by the Earth's surface and atmosphere in the NIR and

SWIR spectral bands (Hu et al., 2018). Recent studies show the potential of using high-resolution TROPOMI $XCH_4$ for detection and quantification of the $CH_4$ emissions. TROPOMI observations have been used for quantifying $CH_4$ emissions from the oil and gas sector (Pandey et al., 2019; Varon et al., 2019; de Gouw et al., 2020; Schneising et al., 2020; Zhang et al., 2020), from urban areas (Tu et al., 2022a; Foy et al., 2023; Plant et al., 2022), and from coal mining (Sadavarte et al., 2021; Tu et al., 2022b). In this study, the TROPOMI $XCH_4$ observations spanning the period from May 2018 to May 2023 over the

study areas in the Shanxi province are used. A data quality filter (qa = 1.0) is applied to characterize the data during clear-sky and low-cloud atmospheric conditions.

## 2.2 CH$_4$ inventory datasets

Qin et al. (2024) used both public and private datasets from over 600 individual coal mines in Shanxi Province. The IPCC Tier 2 approach is applied to calculate the corresponding CH$_4$ emissions based on 3-5 sets of observed emission factors, thereby establishing a range of bottom-up estimation of CMM on a mine-by-mine basis. In the following work, the bottom-up inventory computed from the median emission factors (E5) will serve as a prior information in the wind-assigned method for estimating emissions, referring to IPCC Tier 2 bottom-up inventory. In their study, an eddy-covariance tower was installed in Changzhi during two two-month periods to derive an average observed CH$_4$ flux. Based on the in-situ measurements, a series of scaling factors at different percentiles of the observational distribution (i.e., 10%, 30%, 50%, 70%, 90%) were generated. These scaling factors were subsequently employed to update the preliminary Tier 2 bottom-up inventory (Qin et al., 2024). The scaling factors for a specific percentile of the observational distribution show minimal variations among different coal mines, suggesting these factors can be treated as constant values across the ensemble of coal mines at each percentile. Our wind-assigned method emphasizes the proportional share of emissions per mine rather than absolute values, resulting in estimated CMM emissions that do not significantly differ whether using the Tier 2 bottom-up inventory or one of the scaled inventory datasets. In additional to the current IPCC 2 Tier bottom-up inventory, the scaled inventory is also provided as an additional reference point in this work.

The CAMS Global anthropogenic emissions (CAMS-GLOB-ANT) inventory provides methane emissions for different sectors with a spatial resolution of 0.1º × 0.1º and temporal coverage from 2000 to 2024 (Granier et al., 2019; https://permalink.aeris-data.fr/CAMS-GLOB-ANT, last access: 12 July 2023). Emissions are provided as monthly and yearly averages and v5.3, which includes updated ship emissions from CAMS-GLOB-SHIP v3.1, is used in this study. The yearly mean of CAMS-GLOB-ANT for 14 sectors are illustrated in Figure 1a. Emissions from maritime transport in the study area are zero and not shown here. The inventory very well presents the dominant emission sources in the study area. The coal production (fugitives (coal)) is the dominant source of methane emissions, accounting for ~96% of the total emissions. The sector of solid waste and waste water is the second most important emission source which contributes to 2%. Figure 1b shows the spatial distribution of coal emission in Changzhi and the corresponding distributions for Jincheng and Yangquan are presented in Figure A- 1. The locations of the coal mines and the corresponding emission rates are in good agreement with the CAMS inventory. About half of the coal mines are concentrated in the southern region, while the other half are located further north, along a southwest to northeast direction. The two reddish grid points (36.05ºN-36.15ºN, 113.05ºE) denote the highest emission rate in the CAMS inventory, partly due to the Changzhi city, which is located nearby. The CH$_4$ emission in the city region are primarily attributed to the traffic (particularly during the morning and evening rush hours), CH$_4$ leaks at gas stations and is released by the utilization and release of natural gas in residential areas (Liu et al., 2022). The CH$_4$ emission accounts for $1.77 \times 10^{27}$ molec. s$^{-1}$ (1.5 Tg yr$^{-1}$) for the whole study area, which is 55% higher than the IPCC Tier 2 bottom-up inventory ($1.14 \times 10^{27}$ molec. s$^{-1}$ ~ 0.96 Tg yr$^{-1}$).

The EDGARv7.0 emission inventory is the first product of the new Emissions Database for Global Atmospheric Research (EDGAR) Community GHG emissions database (Crippa et al., 2021), which provides estimates of emissions of the three main GHGs ($CO_2$, $CH_4$ and $N_2O$) and fluorinated gases per sector and country. The dataset offers the same spatial resolution of 0.1º × 0.1º as the CAMS inventory and covers the period of 1970 to 2022. The $CH_4$ emissions from the fuel exploitation sector are the dominant $CH_4$ sources in the study area, accounting for 95.5% of the total $CH_4$ emissions during 2018-2021 (Figure A- 2 left). The total estimates originating from the energy sector are around $1.85 \times 10^{27}$ molec. s-1 (1.6 Tg yr-1). The EDGARv7 estimates a very similar spatial distribution (Figure A- 2 right) as the CAMS inventory with slightly higher (4.5%) values in Changzhi. The spatial patterns in Jincheng and Yangquan are presented in Figure A- 3.

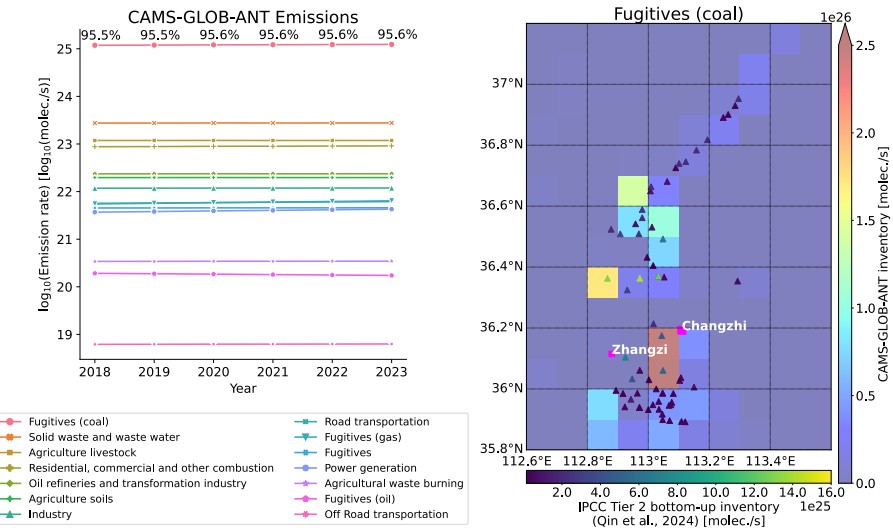

**Figure 1: Left: time-series plot of the yearly averaged CAMS global anthropogenic emissions for different sectors for 2018–2023 (https://permalink.aeris-data.fr/CAMS-GLOB-ANT, last access: 12 July 2023, Granier et al., 2019). The percentage values represent the share of methane emission from coal production and distribution (fugitives (coal)). Right: spatial distribution of methane emission from coal production for the CAMS-GLOB-ANT inventory. The triangle symbols denote the locations of the coal mines and the respective colors represent their emission rates based on the IPCC Tier 2 bottom-up inventory .**

## 2.3 Dispersion model

### 2.3.1 Cone plume model

The $CH_4$ emerging from a point source is expected to be distributed along the wind direction. It is assumed that the $CH_4$ molecules disperse evenly along a fan-shaped plume (Tu et al., 2022a). The column enhancement in the downwind side due to the assumed source is represented by the following equation:

$$dcol_{CH_4} = \frac{\varepsilon}{v \cdot d \cdot fov}$$    **Eq. (1)**

wherein $\varepsilon$ represents the emission rate at the source point in molec. s-1, $v$ is the wind speed in ms-1 with the ERA5 wind data at 100 m employed in this study, $d$ the distance between the source point and the downwind point, and $fov$ the opening angle of the cone plume in rad. Here $fov$ is assumed to be 60º based on previous studies (Tu et al., 2022a, b). It should be noted that

the point source does not generate enhanced CH4 concentrations outside of the cone. This may introduce some uncertainties and will be discussed in Sect. 3.4.

### 2.3.2 Gaussian plume model

The dispersion of a gas as a function of distance downwind from a point source can alternatively be approximated by a
Gaussian plume model (Seinfeld and Pandis, 2006). To evaluate the sensitivity of the analysis with respect to the cone plume model assumption, as an alternative a Gaussian plume model is investigated in the following.

$$dcol_{CH_4} = \frac{\varepsilon}{v \cdot d \cdot \sqrt{2\pi} \cdot (\frac{fov}{2})} \cdot \exp\left(-\frac{1}{2} \cdot (\frac{\varphi}{(\frac{fov}{2})})^2\right) \qquad \textbf{Eq. (2)}$$

where $dcol_{CH_4}$ represents the enhanced column in the downwind direction, the $fov$ the angle of the opening angle adopted from the cone plume model (Tu et al., 2022a), $v$ the wind direction, $d$ the distance between the point source and the downwind location, and $\varphi$ the angle of plume axis and the direction under consideration. The $dcol_{CH_4}$ in the cone plume is restricted in
the cone area with an opening angle of $fov$, while the values in the Gaussian plume show a gradually fading enhancement along the circle arc at any radius d (Figure A- 4). We use the Gaussian plume model here in addition to the cone plume outlined before for enhancing the error estimate of the emissions resulting from our inversions. We estimate the error budget by varying the model parameters of each model description within reasonable limits. Using two alternative models for describing the gas dispersion, in addition enables us to investigate the uncertainties introduced by the chosen model type.

**2.4 Background removal and wind-assigned anomaly method**

It is of importance to separate the increase of the atmospheric CH4 concentration due to local emissions from the accumulated atmospheric CH4 background concentration (the CH4 atmospheric lifetime is in the order of 12 years). A Jacobian matrix is introduced to reconstruct the background according to a few background model coefficients, i.e., a constant CH4 value and superimposed disturbances: a temporal linear increase, a seasonal cycle determined by the amplitude and phase of the three
frequencies 1/year, 2/year and 3/year, a daily signal (same value for all data measured during a single day), and a horizonal gradient (same value for any time but dependent on the horizontal location) (Tu et al., 2022a). In the following discussion, the satellite enhancements refer to the residual signal as deduced from TROPOMI CH4 observations after subtracting the modelled background (Figure 4 lower panel).

The wind-assigned anomaly method was first developed for quantifying CH4 emissions from landfills in Madrid (Tu et al.,
2022a). Its applicability for estimating the CMM emissions in the Upper Silesian Coal Basin (USCB) in southern Poland was demonstrated afterwards (Tu et al., 2022b). The wind-assigned anomalies refer to the difference of enhancements under two opposite wind regimes ((e.g., NW (>215º and <45º) and SE (45º – 215º) fields for Changzhi region). The wind regimes are divided mainly based on the predominant wind fields over the study regions. The expected daily enhancements (plumes) generated by individual emission sources are computed based on Eq. (1) and all contributions then are superimposed to obtain

a total daily pattern of CH₄ enhancement due to local sources. A temporally averaged pattern is obtained for each wind regime over the study period and the difference between the two patterns is therefore the modeled wind-assigned anomaly. The empirical anomalies are computed from the satellite XCH₄ data. The estimated emission rate is computed by scaling the modeled anomalies to the empirical anomalies. The uncertainties of the empirical anomalies are determined by the deficits of the background model resulting in an imperfect elimination of the background, and the noise errors in the satellite observations.

## 3. Results and discussion

### 3.1 TROPOMI observations


Shanxi province is rich in coal resources, and as a result, there are more than 600 coal mines spread across the province. Most of these coal mines are concentrated in the northern, eastern and southeastern, and central regions of Shanxi. A multi-year average of TROPOMI XCH₄ observations in the whole Shanxi province is shown in Figure 2, superimposed to the locations of mines in the area. Elevated XCH₄ is observed in three regions: Yangquan (east), Changzhi (southeast) and Jincheng (south). Of these regions, the Changzhi region is of particular interest since a field campaign was implemented in 2022. This field campaign region covers an area of 35.8ºN–37.2ºN, 112.6ºE–113.6ºE, i.e., 155 km × 90 km) and will be discussed in detail as an example to better understand the CH₄ emissions from coal mining activities in Shanxi province.


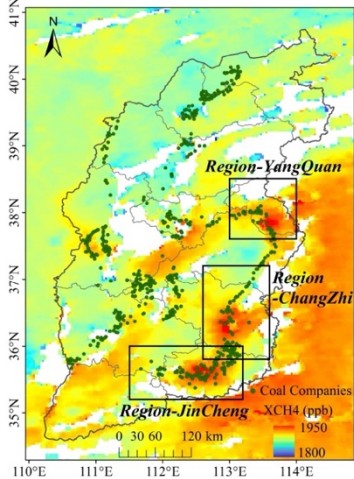

**Figure 2: TROPOMI XCH₄ and the location of coal mines in Shanxi province. Green dot symbols denote the coal mine locations (http://nyj.shanxi.gov.cn/, last access: August 21, 2023).**

There are 62 coal mines located over the study area in Changzhi region, as shown in Figure 3. The emission rates range from $1.6 \times 10^{24}$ to $1.4 \times 10^{26}$ molec. s⁻¹ (~0.001 Tg yr⁻¹ – 0.11 Tg yr⁻¹) (Qin et al., 2024). There are near 30 small coal mines scattered in the mountain area in the south and each mine has a relatively low emission rate, measuring less than $1.0 \times 10^{25}$

molec. s⁻¹. Some larger coal mines with higher emissions rates (emission rate > $1.0 \times 10^{25}$ molec. s⁻¹) are found close to the

Zhangzi county as well as in the north region and the mean value for these is around $7.3 \times 10^{25}$ molec. s$^{-1}$ with a standard deviation of $4.6 \times 10^{25}$ molec. s$^{-1}$.

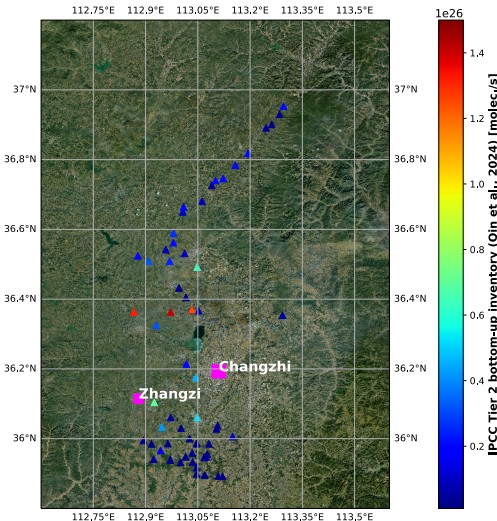

**Figure 3: Terrain map with IPCC Tier 2 bottom-up inventory (Qin et al., 2024). The triangle symbols represent the location of all individual coal mines, and different colors denote the emission rates. The square symbols denote the locations of Changzhi city and Zhangzi county. Terrain information originates from World Imagery.**

A time series of five-years of TROPOMI XCH$_4$ observations in the Changzhi region is shown in Figure 4. The average concentration is $1906.8 \pm 41.0$ ppb over the entire period. From 2019 to 2022, there is an observed increase in XCH$_4$ levels by approximately 0.7% per year. The observations in the figure indicate that there is a clear seasonal variability in the concentrations. The data show that the lowest abundances of XCH$_4$ occur in the early part of the year, while the highest values are observed in autumn. The seasonal pattern is determined by both sinks and sources. The elimination of methane (CH$_4$) by hydroxyl radicals (OH) in the troposphere, known as atmospheric oxidation, plays a crucial role in controlling the concentrations of climate-relevant gases like CH$_4$ (Rigby et al., 2017; Li et al., 2018). This process is responsible for approximately 85-90% of atmospheric CH$_4$ loss (Saunois et al., 2020). On the other hand, the dominant factor contributing to CH$_4$ emissions in this region is coal mining activities. These coal production activities can vary throughout the year and have a significant impact on the overall XCH$_4$ concentrations.

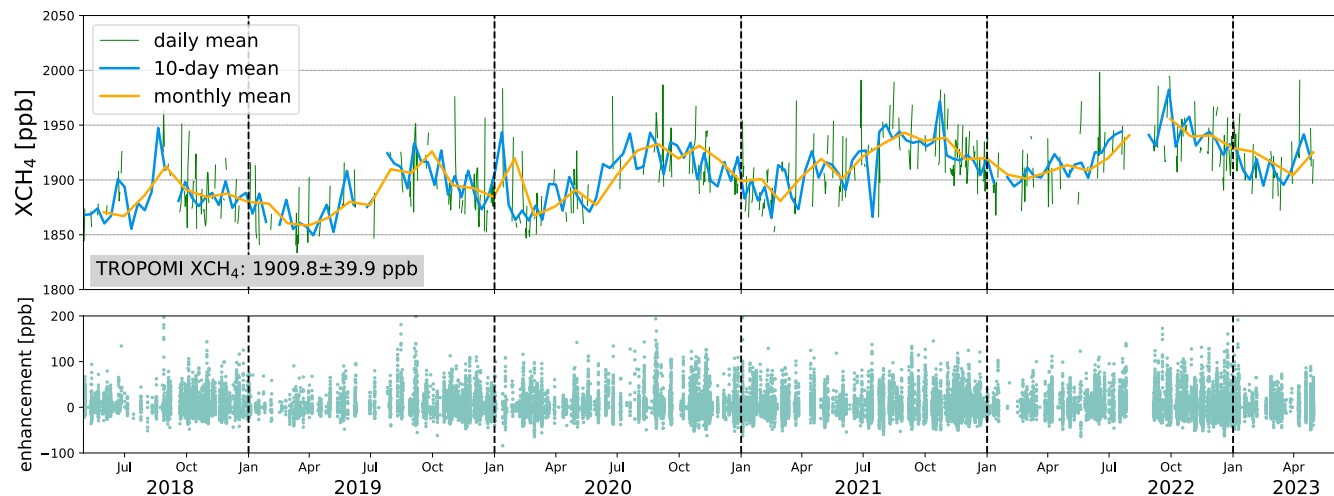

**Figure 4:** Time-series of average TROPOMI XCH$_4$ (upper panel) and corresponding enhancement after removing the background (lower panel) over Changzhi region (35.8ºN—37.2ºN, 112.6ºE—113.6ºE) from May 2018 to April 2023.

## 3.2 Estimation of CH$_4$ emission strengths from TROPOMI data sets

The ERA5 wind at 100 m altitude above ground is used for describing the transport with the wind-assigned method. The wind is segmented as NW (>215º and <45º) and SE (45º – 215º) fields for Changzhi region (Figure A- 5). Due to the observed seasonal changes in XCH$_4$, the observed variable background concentrations need to be considered when estimating the emissions. The TROPOMI enhancements after removing the background are shown in Figure 5a. High values are observed in the center and south of the study area, i.e., close to the clusters of the coal mines (triangle symbols). It is difficult to distinguish the CH$_4$ from the residential regions since the coal mines are located close to Zhangzi county and Changzhi city. The averaged enhancements are 4.7 ppb ± 5.6 ppb for the whole region. The wind-assigned anomalies from the TROPOMI observations indicate the difference of the enhancements for wind coming from NW and from SE, resulting in a positive plume in the SE direction and negative plume in NW direction (Figure 5b).

The correlation of the wind-assigned anomalies deduced from the TROPOMI observations and from the plume model using the IPCC Tier 2 bottom-up inventory (Qin et al., 2024) is presented in Figure 5c, and the estimated emission rate is 8.4 × 10$^{26}$ molec. s$^{-1}$ (R$^2$ = 0.61). In comparison to our results based on the TROPOMI observations and the wind-assigned method, the IPCC Tier 2 bottom-up (Qin et al., 2024), CAMS and EDGARv7 inventories are overestimating the emissions, and have a high bias by 31%, 120% and 130%.

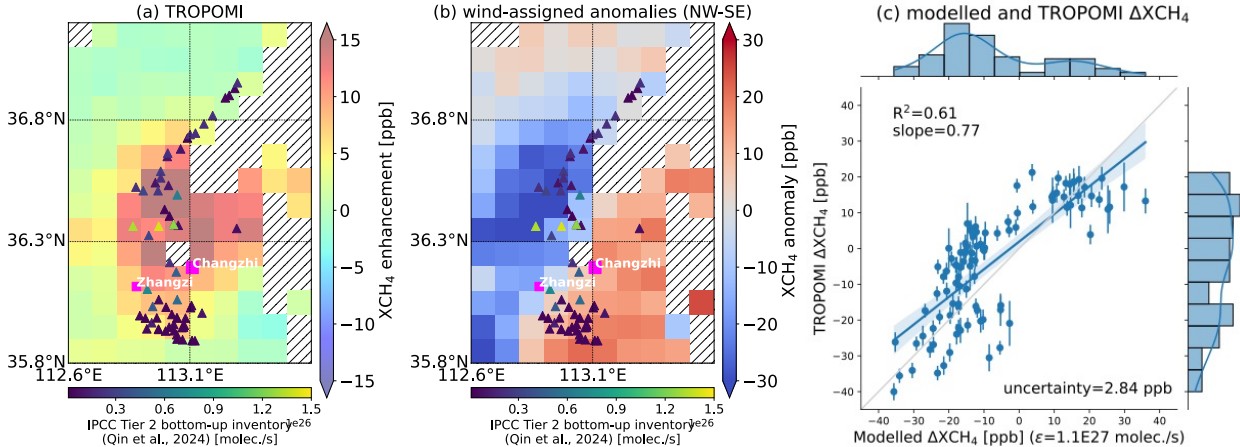

**Figure 5:** TROPOMI $XCH_4$ enhancement ($XCH_4$-background) (a), the wind-assigned anomalies (NW–SE) (b), and correlation plot of the wind-assigned anomalies (c) between TROPOMI and the simple cone plume model with using the IPCC Tier 2 bottom-up inventory ($1.1 \times 10^{27}$ molec. $s^{-1}$ in total, Qin et al., 2024) and ERA5 wind at 100 m during May 2018–May 2023 over the Changzhi region. The triangle symbols denote the inventory locations, with different colours indicating varying emission rates. Hatched areas in (a)–(b) indicate grids with no available data. The uncertainty in (c) is presented by the average error bars of the anomalies, which are derived from the uncertainty in the background and the TROPOMI observations.

The wind-assigned method was also applied to Jincheng and Yangquan regions. The wind segmentations are NW-SE for Jincheng and E-W for Yangquan based on the ERA5 wind information (Figure A- 5). The estimated emission is $1.4 \times 10^{27}$ molec. $s^{-1}$ for Jincheng and $4.9 \times 10^{26}$ molec. $s^{-1}$ for Yangquan. The wind-assigned anomalies in the Jincheng region shows a better correlation with a $R^2$ value of 0.80, whereas the value is lower ($R^2$=0.42) in the Yangquan region (Figure A- 6). The resulting estimate for Jincheng is close to the IPCC Tier 2 bottom-up inventory (Qin et al., 2024), displaying a minor deviation of around 8%. However, the distinction is more pronounced for Yangquan, exhibiting a slightly larger difference of 14%. Figure 6 summarizes the estimated emissions based on the wind-assigned anomaly method compared to the predictions based on the inventories in all regions. In general, the estimates are comparable to the IPCC Tier 2 bottom-up inventory (Qin et al., 2024), whereas both CAMS and EDGAR inventories overestimate the emissions with a relative difference of about 120%/130% in Changzhi, 60%/68% in Jincheng and 165%/186% in Yangquan.

Our CMM estimates in these three regions fall within the 30[th] and 70[th] percentile range of the updated emission rates in the study by Qin et al. (2024). In addition, our results are consistently lower than the CAMS-GLOB-ANT and the EDGARv7 inventories. This result agrees with previous studies. For instance, a -15% underestimation compared to the UNFCCC has been reported by Chen et al., (2022). Additionally, Zhang et al., (2021) documented a 30% decrease in their posterior estimates for China, with 60% attributed to coal mining. This pattern of overestimated anthropogenic emissions, in comparison to China's inventory, has also been found in previous research, utilizing GOSAT inversion and various versions of the EDGAR inventory as a priori estimates (Miller et al., 2019; Maasakkers et al., 2019). This divergence may be attributed to two reasons: (1) missing observation of strong CMM emissions during the TROPOMI overpass. It is important to note that CMM emissions exhibit a strong dependency on coal mine activities, which vary over time. The TROPOMI data provide instantaneous observations,

capturing CH$_4$ concentrations at a specific moment (local time ~ 13:30), thereby leading to limitations in detecting strong CMM emissions during both morning and afternoon periods. (2) the CMM utilization connected with a reduction of release into the atmosphere has been largely improved in the last decade, since the national government issued specific targets in the national 12$^{th}$ and 13$^{th}$ five-year plan (Gao et al., 2021; Lu et al., 2021).

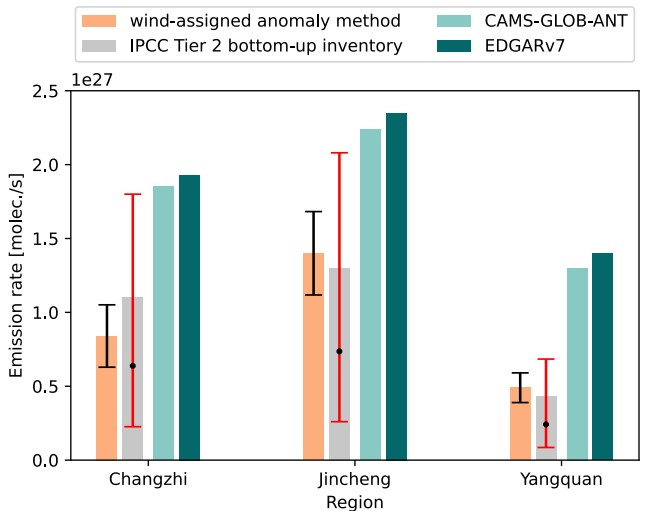

260

**Figure 6: Estimated emission rates and emission rates from three different inventories for the Changzhi, Jincheng and Yangquan regions. The dot symbols in the grey bars represent the emission rates updated with the flux tower observations with 50$^{th}$ percentile of distribution, and the bottom and top error bars (red) represent the values with 30$^{th}$ and 70$^{th}$ percentile, respectively (Qin et al., 2024).**

265 **3.3 Uncertainty analysis**

**3.3.1 Background removal**

In comparison to the atmospheric concentration, the CH$_4$ amounts emitted from the sources are relatively small (Figure 4). To assess the impact of background removal sensitivity, the 10$^{th}$ lower percentile of overall satellite observations each day is considered as the background for the study area on that day, instead of separately considering the spatial and temporal variation
270 as described in Section 2.3. The XCH$_4$ enhancements using the new background removal method, are generally higher than those achieved with the previous approach, exhibiting a mean bias of 21.5 ± 14.4 ppb in Changzhi (Figure A- 7a). This discrepancy diminishes to -3.6 ± 2.1 ppb when comparing the wind-assigned anomalies computed from TROPOMI enhancements based on different background removal methods (Figure A- 7b). Calculating the differences in enhancements under two different wind field segmentations helps to reduce systematic errors associated with the background removal. The
275 substitution of the background removal method results in a 7% increase in estimated emission rates in Changzhi, a 6% increase in Jincheng and a 9% increase in Yangquan.

### 3.3.2 Cone plume and Gaussian plume model

To further investigate the uncertainty of the two plume models, different opening angles are tested for estimating $CH_4$ emissions in the study regions. Estimated emissions increased with increasing *fov* for both plume models (Figure 7 for Changzhi, and Figure A- 8 for Jincheng and Yangquan). The results based on the Gaussian plume is higher than those based on the cone plume and the discrepancy between the two models increases with increasing opening angle. For Changzhi region, when the *fov* is chosen as the previous setting value (60º), the estimated emission rate based on the Gaussian plume are $9.4 \times 10^{26}$, which is 12% higher that based on the cone plume model. The relative difference between these two models drops to 5% for *fov* = 20º. The anomalies derived from the Gaussian plume model are overall similar to those from the cone plume model, showing a slightly better correlation with the anomalies from the TROPOMI observations ($R^2 = 0.65$, Figure A- 9).

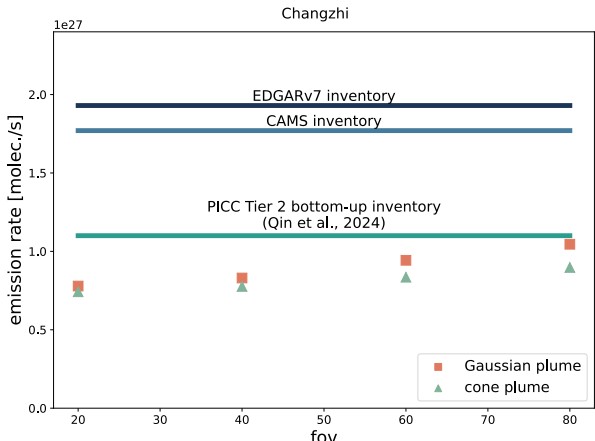

**Figure 7: Estimates of emission rates in Changzhi region with respect to different opening angles based on cone plume and Gaussian plume model. The three different inventories are presented as well.**

### 3.3.3 Wind analysis data and field segmentation

Uncertainty in wind direction and speed is one of the largest sources of error in correctly estimating the emission rates (Tu et al., 2022b). Thus, use of winds at different model height level and different wind field segmentations are tested, the spatial variation of the winds is investigated, and an alternative wind data set is applied for the wind-assigned anomaly analysis.

The wind direction exhibits a similar pattern at 10 m and 100 m model levels, while the speed increases with height. The wind speed at 10 m is 15.4% lower than that at 100 m in Changzhi region. Consequently, the corresponding estimated emission rate amounts to $7.4 \times 10^{26}$ molec. s$^{-1}$, representing an 11.9% decrease compared to the estimate obtained using wind data at 100 m. Using 10 m winds instead of 100 m reduces the emission estimates by 10.7% (the average wind speed reduces by 17%) in Jincheng and 4.1% (the average wind speed reduces by 10%) in Yangquan for wind at 10 m, in comparison to those using wind data at 100 m. The wind segmentation is mainly based on the local predominant wind regimes. To quantify its uncertainty, different segmentations (N and S segmentations for Changzhi and Jingcheng, and NW and SE segmentations for Yangquan)

are applied. The corresponding $CH_4$ emission increases by 12% in Changzhi, whereas it decreases by 7% in Jincheng and by 6% in Yangquan.

    The topography can affect the surface wind, making it challenging to determine the exact pathway of transport from the emission source to the measurement station (Babenhauserheide et al., 2020). Considering the elevation features, the central part of the Changzhi region is characterized by flat terrain, while elevations rise in the northeast and southeast, as depicted in

Figure A- 10. To further investigate the sensitivity of the wind spatial variation, the wind data at the central point (36.5º N, 113º E for Changzhi, 35.5º N, 112.75º E for Jincheng and 38º N, 113.5º E in Yangquan) is used as a representative value to represent the wind for the entire study area. The wind direction pattern at the central point tends to have more wind from east and the averaged wind speed decreases only by 4.4% compared to that over the whole study area in Changzhi (Figure A- 11). This substitution results in a decreased estimated emission rate with 11% (emission rate: $7.5 \times 10^{26}$ molec. s$^{-1}$) in Changzhi,

an increased emission rate with 7% ($1.5 \times 10^{27}$ molec. s$^{-1}$) in Jincheng and 8% ($5.3 \times 10^{26}$ molec. s$^{-1}$) in Yangquan.

    An alternative wind dataset at a height of 100 m above the ground from the Global Data Assimilation System (GDAS) are used instead of the ERA5 dataset. The GDAS FNL (Final) operational global analysis and forecast data is provided by the National Centers for Environmental Prediction (NCEP) with a spatial resolution of 0.25º × 0.25º and a temporal resolution of 6 h (National Centers for Environmental Prediction et al., 2015). Compared with the ERA5 data, the NCEP data shows

comparable wind distributions (Figure A- 13), featuring slightly elevated wind speeds and a more prevalent wind direction originating from the broader northwest region. The estimated emission strength for using the NCEP dataset amounts to $7.9 \times 10^{26}$ molec. s$^{-1}$, indicating a 6% reduction compared to the value obtained using the ERA5 dataset in Changzhi. The estimates increase by 7% in Jincheng and decrease by 2% in Yangquan.

### 3.3.4 Inventories

The use of an inventory as a priori knowledge is integral to the wind-assigned anomaly approach. As detailed in Sect. 3.2, the IPCC Tier 2 bottom-up and the CAMS-GLOB-ANT (or EDGAR) inventories highlight discrepancies in emission sources both in terms of location and abundance. To investigate the uncertainty introduced by the inventory, the CAMS-GLOB-ANT inventory is employed in the approach instead of the IPCC Tier 2 bottom-up inventory. Given the similarity in pattern between the CAMS and EDGAR inventories, we have exclusively focused on the CAMS inventory. This substitution leads to minor

deviations in the observed enhancements (TROPOMI XCH$_4$ - background) across the Changzhi region (Figure A- 14(a) and Figure 5(a)). In general, the spatial patterns maintain a notable similarity, while presenting some divergence in abundance. The calculated average stands at 3.61 ± 4.44 ppb when employing the CAMS-GLOB-ANT inventory. In contrast, using the IPCC Tier 2 bottom-up inventory yields an average of 4.68 ± 5.59 ppb. These two datasets show a mean bias of 1.12 (± 2.93) ppb with an $R^2$ value of 0.8562 (Figure A- 14(b)). The wind-assigned anomalies from both datasets also present comparable

patterns and display a strong correlation between them (Figure A- 14(c), $R^2 = 0.9962$). It is because the systematic errors in background removal is compensated by computing the differences of enhancements under different wind field segmentations. The estimated emission strength amounts to $8.5 \times 10^{26}$ molec. s$^{-1}$ in Changzhi, which is very close (1%) to the strength

estimated using the Tier 2 IPCC bottom-up inventory as the a priori. Similar results are observed in the other two regions, with biases of 7% higher and 12% lower biases in Jincheng and Yangquan, respectively.

Based on the error propagation, the total uncertainties in the estimated emission rates from the different error sources (background removal and noise in the satellite data, dispersion model (Gaussian plume and opening angle $fov$ = 70º), wind information (ERA5 wind for height level = 10 m, wind without considering spatial variation, different wind segmentation, and NCEP wind data), and different inventories) are approximately 25% for Changzhi, 20% for Jincheng and 21% for Yangquan.

## 4. Conclusion

Quantifying CMM emissions using high-spatial resolution satellite observations can contribute independent emission estimates for evaluating inventories and assisting in the development of reduction strategies and interventions. In this study, a wind-assigned anomaly method was used for analyzing the TROPOMI XCH$_4$ observations between May 2018 to May 2023. The CMM emissions in three subregions (Changzhi, Jincheng, Yangquan) in the coal-rich Shanxi province of China were achieved. The three regions are aggregation areas of coal mines, consequently exhibiting elevated XCH$_4$ abundances. The

concluded emission strengths are 8.4× $10^{26}$ molec. s$^{-1}$ (0.706 Tg yr$^{-1}$, ± 25%), 1.4 × $10^{27}$ molec. s$^{-1}$ (1.176 Tg yr$^{-1}$, ± 20%), and 4.9 × $10^{26}$ molec. s$^{-1}$ (0.412 Tg yr$^{-1}$, ± 21%), respectively.

    The estimates obtained derived through the wind-assigned anomaly method demonstrate comparability with the IPCC Tier 2 bottom-up inventory (Qin et al., 2024). Compared to the estimates, the inventory shows relative differences of 31%, -7%, and -12% in Changzhi, Jincheng, and Yangquan, respectively. Our CMM estimates in these three regions fall within the 30[th]

and 70[th] percentile range of the updated emission rates in the study by Qin et al. (2024). The CAMS-GLOB-ANT and EDGARv7.0 inventories show very similar results. However, higher discrepancies are found when comparing our estimates to these inventories, with differences reaching approximately 125%, 64%, and 176%, respectively. This indicates a potential overestimation of CH$_4$ emissions from these coal mining regions in the CAMS-GLOB-ANT and EDGARv7.0 inventories. Previous studies have also documented similar trends, reporting overestimated CMM emission estimates in inventories (Chen

et al., 2022; Zhang et al., 2021; Miller et al., 2019; Maasakkers et al., 2019). In additional, our lower estimates might due to two reasons: (1) a lack of observation of strong CMM emissions during TROPOMI overpasses. CMM emissions are closely tied to coal mine activities, which exhibit temporal variability. TROPOMI data provide instantaneous observations, capturing CH$_4$ concentrations at a specific local time (~ 13:30), which may limit the detection of strong CMM emissions during morning and afternoon periods. (2) improvement in CMM and reduction of atmospheric release have been substantial in the last decade.

This improvement is attributed to specific targets set by the national government in the national 12[th] and 13[th] five-year plan (Gao et al., 2021; Lu et al., 2021), indicating a potential decrease in actual emissions compared to historical inventory estimates.

    To evaluate uncertainties, we explore variations in background removal, the dispersion model and its inputs (wind data and inventory serving as a priori knowledge). The background removal introduces upward biases in 6%-9% when using the 10[th] lower percentile of overall satellite observations each day as the background for the study area on that day. The cone plume

model introduces uncertainties due to assuming a sharply bordered fan-shaped plume extending along the downwind direction, i.e., any points located outside of the cone area experience no enhancement. To estimate uncertainties connected to the assumed plume shape, we investigate the assumption of a Gaussian plume, resulting in an estimated emission strength increase of 12%, 7% and 8% in Changzhi, Jincheng, and Yangquan, respectively. Beyond consideration of the dispersion model, the assumed wind speed and direction represent major sources of uncertainty. An analysis of using wind at 10 m height reveals lower biases of 4%-12%. Additionally, we tested an alternative wind category (N-S), yielding a 10% increase in estimated emission strength in Changzhi and 7% and 6% decrease in Jincheng and Yangquan (NW-SW), respectively. Considering the elevation features, the spatial variation in wind leads to median biases ranging from 6% to 9%. Introducing another wind dataset (NCEP FNL) for analysis results in different biases of -6%, 7% and -2% in the three different regions. The emission inventory is considered as a priori knowledge in the approach and replacing the IPCC Tier 2 bottom-up inventory with the CAMS-GLOB-ANT inventory introduces small biases (1%, 7% and -12%). Considering all the impacts mentioned above, the total uncertainties (at 1-sigma level), computed through error propagation, are determined to be 25% in Changzhi, 20% in Jincheng and 21% in Yangquan.

This study further demonstrates the practicality of employing the wind-assigned anomaly method together with the high spatial resolution TROPOMI $XCH_4$ to quantify regional-scale $CH_4$ emission strengths. This approach holds promise for extending its application to estimate CMM emission in other coal mine-active regions, thereby providing top-down estimates that can enhance the refinement of inventories. Moreover, these results offer support for enhancement of the mitigation strategies and the efficient control of CMM emissions.

*Data availability*. The TROPOMI data set is publicly available from https://scihub.copernicus.eu/ (last access: 29 July 2023; ESA, 2020). The access and use of any Copernicus Sentinel data available through the Copernicus Open Access Hub are governed by the legal notice on the use of Copernicus Sentinel Data and Service Information, which is given here: https://sentinels.copernicus.eu/documents/247904/690755/Sentinel_Data_Legal_Notice (last access: 29 July 2023; European Commission, 2020).

*Author contributions*. QT and FH developed the research question. QT wrote the manuscript and performed the data analysis with input from FH. QK and JC supplied the IPCC Tier 2 bottom-up inventory and local insights for the study regions. XZ designed and created parts of the graphics. FK and QK participated in result discussions and contributed to improve the paper. All authors contributed to the interpretation of the results and the improvement of the manuscript.

*Competing interests*. At least one of the (co-)authors is a member of the editorial board of Atmospheric Chemistry and Physics.

*Acknowledgements*. We would like to thank Emissions of atmospheric Compounds and Compilation of Ancillary Data
(ECCAD) for providing CAMS-GLOB-ANT inventory data and the Emissions Database for Global Atmospheric Research
(EDGAR) for providing EDGARv7.0 inventory data. Thanks should also go to the TROPOMI team for making CH$_4$ data
publicly available. Furthermore, we extend our appreciation to the two anonymous referees for their valuable comments, which
has improved the final version of this paper.

*Financial support*. This study was supported by the National Natural Science Foundation of China (grant no. 42305138), the
Shanxi Province Major Science and Technique Program (grant no. 202101090301013) and the Fundamental Research Funds
for the Central Universities.

**Appendix**

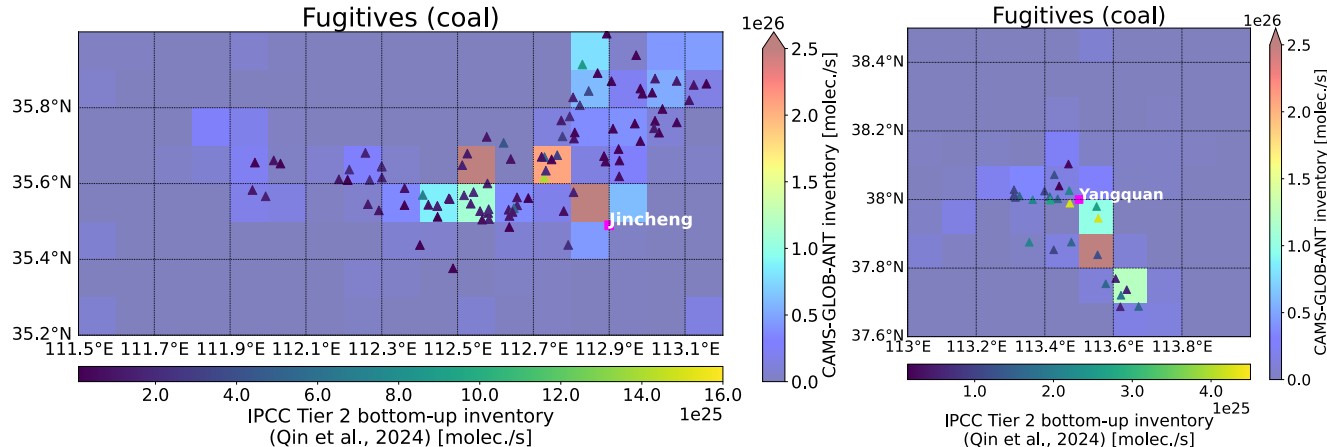

**Figure A- 1: Similar to Figure 1-right, but for Jincheng (left) and Yangquan (right) regions, respectively.**

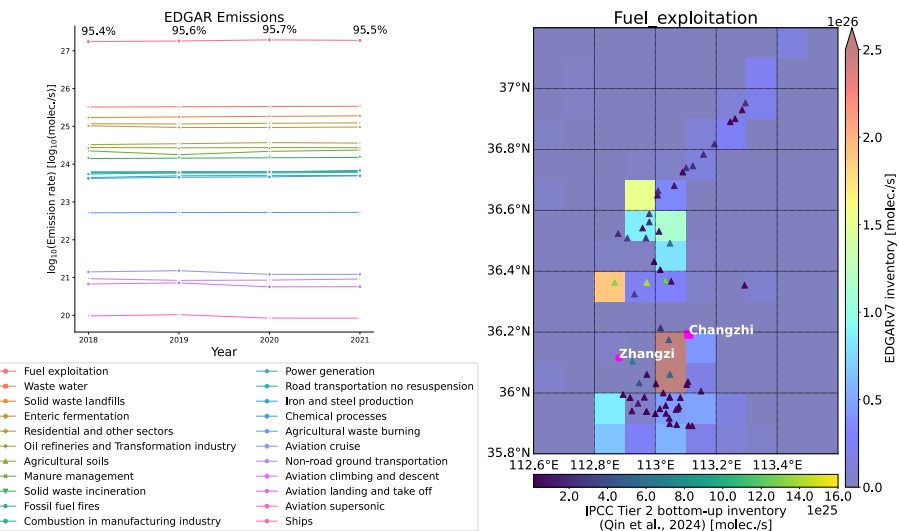

**Figure A- 2: Similar to Figure 1 but for the EDGARv7 energy sector in Changzhi.**

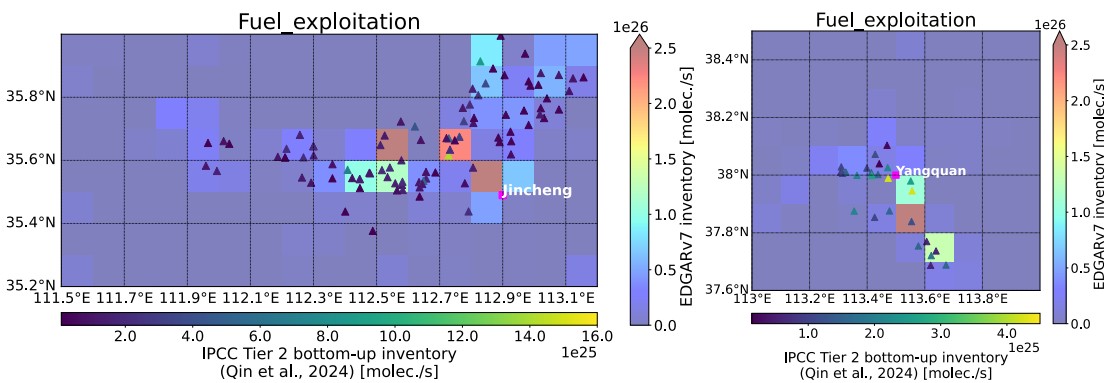

**Figure A- 3: Similar to Figure 1-right, but for the EDGARv7 energy sector in Jincheng (left) and Yangquan (right) regions, respectively.**

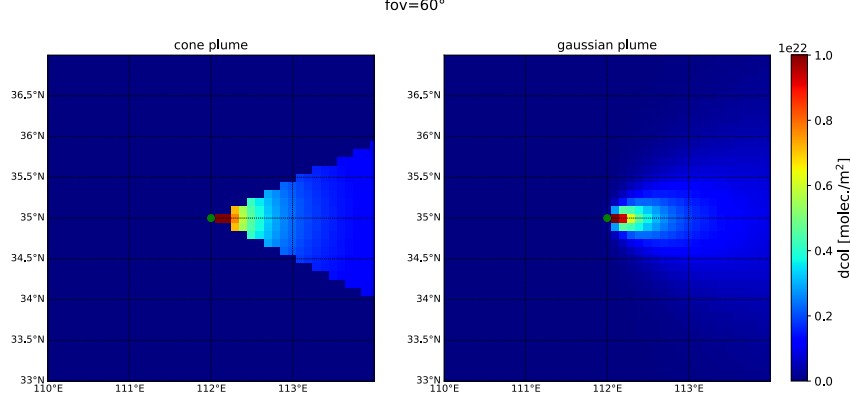

**Figure A- 4: Spatial distribution of dispersion based on the cone plume (left) and Gaussian plume (right) model. The wind from west is used as an example.**

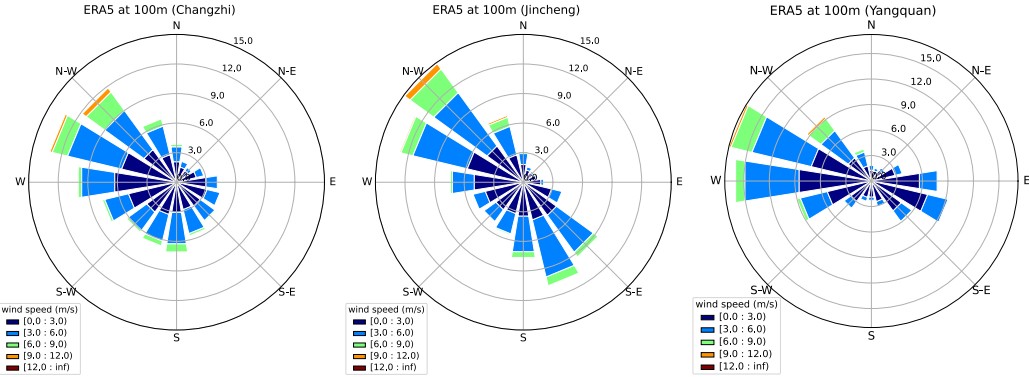


**Figure A- 5: Wind roses plots for local daytime (08:00–18:00 UTC) from May 2018 to April 2023 for the ERA5 model wind in Changzhi, Jincheng and Yangquan regions, respectively.**

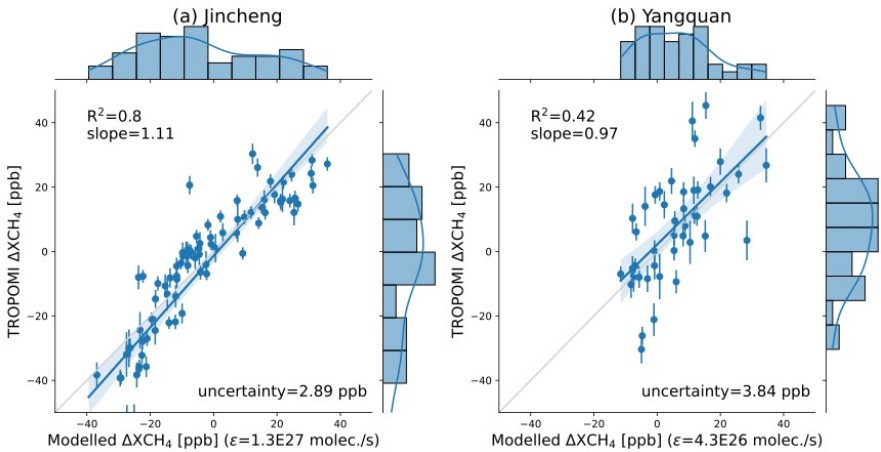

**Figure A- 6: Similar to Figure 5(c), but for the Jincheng and Yangquan regions.**

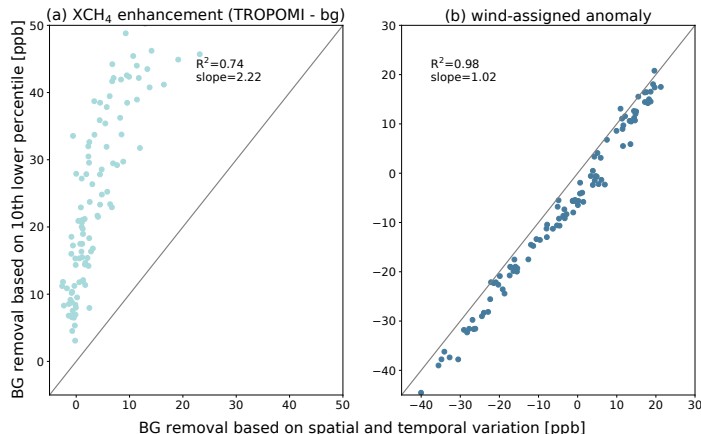


**Figure A- 7: XCH$_4$ enhancements (TROPOMI - background) and its corresponding wind-assigned anomaly using different background removal methods. The grey solid line corresponds to the 1:1 line.**

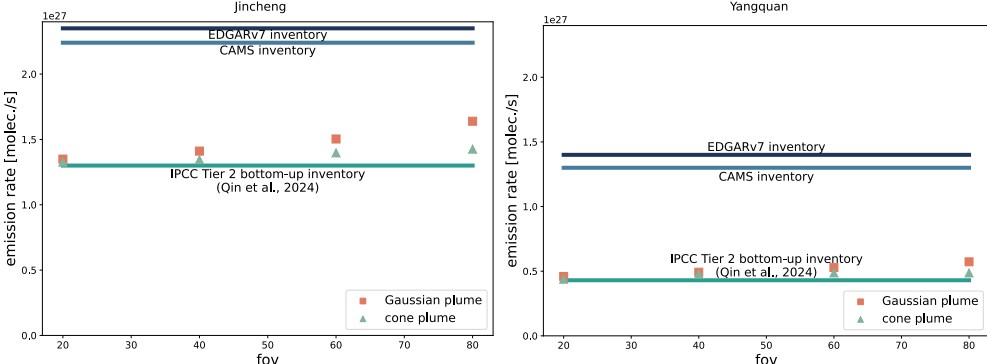

**Figure A- 8: Similar to Figure 7 but for Jincheng and Yangquan regions.**

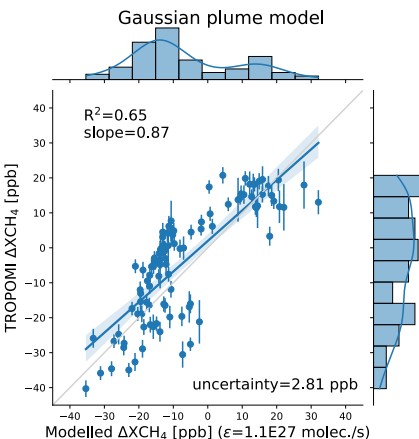


**Figure A- 9: Similar to Figure 5(c) but using the Gaussian plume model.**

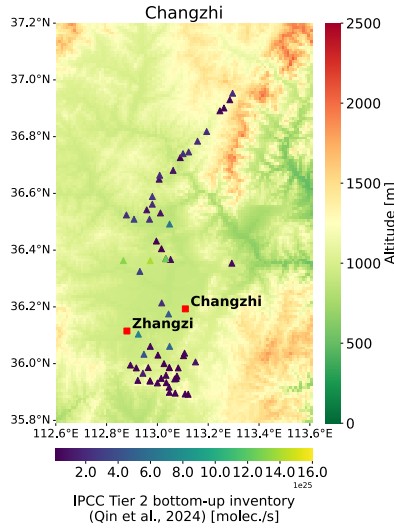

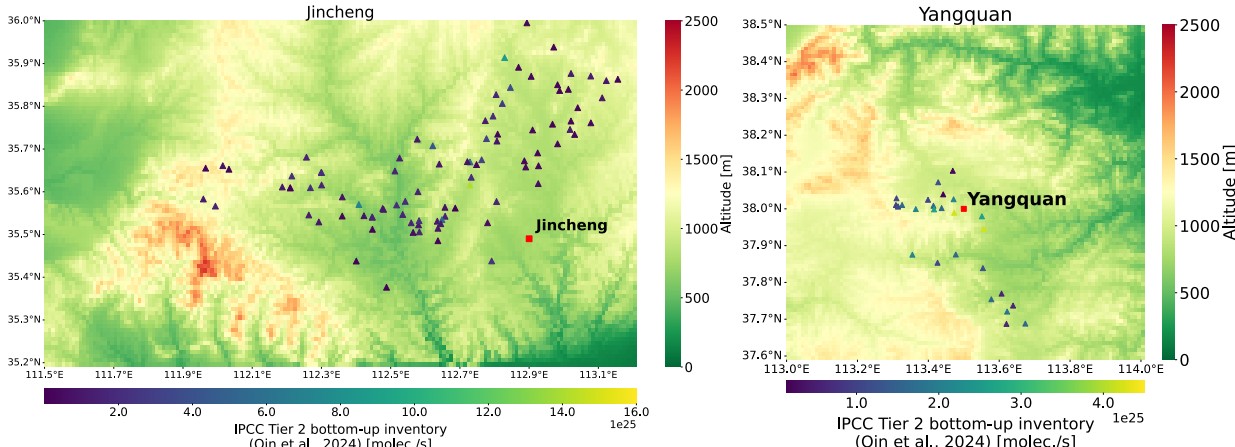

**Figure A- 10: Altitude map for Changzhi, Jincheng and Yangquan regions. Data originate from ALOS World 3D – 30m (AW3D30) (Tadono et al., 2014).**

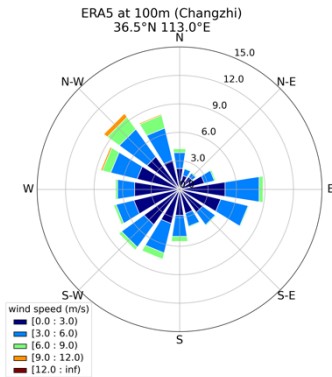

**Figure A- 11: Similar to Figure A- 5(a) but for the grid at 36.5º N, 113º E in Changzhi.**

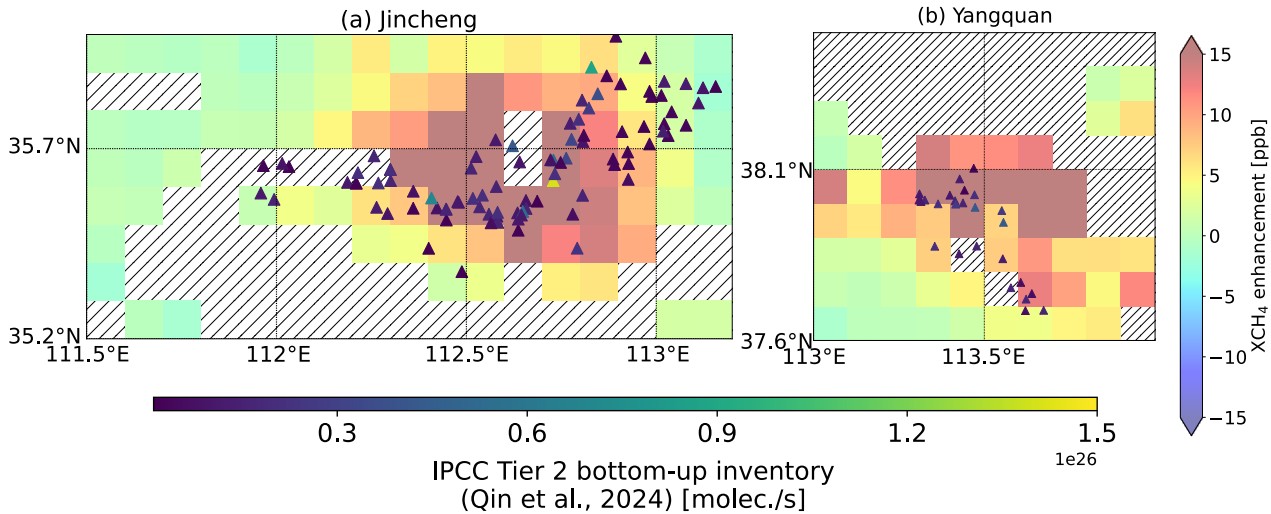

**Figure A- 12: XCH₄ enhancements (XCH₄ - background) for Jincheng and Yangquan region.**

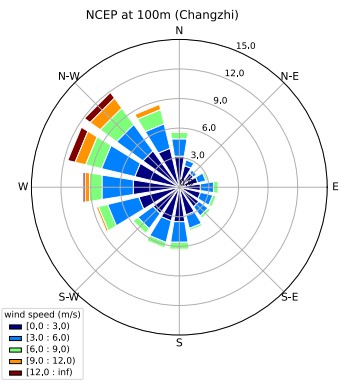

**Figure A- 13: Similar to Figure A- 5(a) but from NCEP FNL operational analysis data in the Changzhi region.**

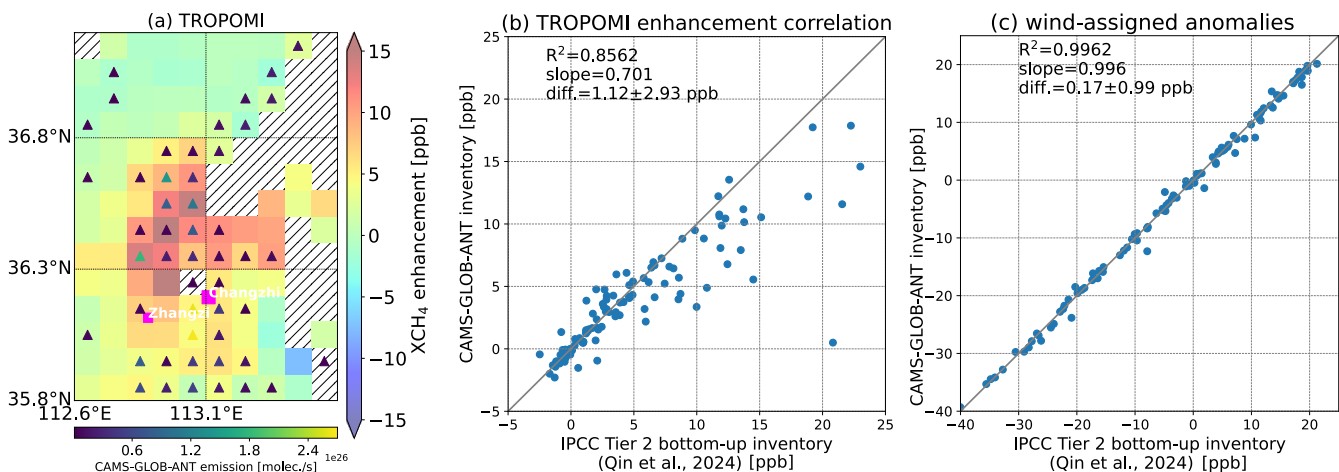

**Figure A- 14: (a): Similar to Figure 5(a) but using the CAMS-GLOB-ANT inventory as a prior information. The triangle symbols denote the inventory location (emission rate > 1 × 10²⁴ molec. s⁻¹). (b): correlation for the enhancement and (c): correlation for the wind-assigned anomalies derived from the TROPOMI observations using the IPCC Tier 2 bottom-up inventory (Qin et al., 2024) and the CAMS-GLOB-ANT inventory. The grey line corresponds to the 1:1 line.**

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
