# Peer review of "Quantifying CH4 emissions from coal mine aggregation areas in Shanxi, China using TROPOMI observations and the wind-assigned anomaly method"

_EGUsphere, 2023_

## Referee Comment (RC1)

This article provides coal mine methane emission estimates for three regions of the Shanxi province (China), using the recently-developed wind-assigned anomaly method with methane concentration observations from the TROPOMI satellite instrument. The results suggest that commonly-used emission inventories overestimate coal mine emissions in the area. The sensitivities of wind-assigned anomaly results to several of the method inputs and parameters are described.

The article is concise (too much actually, see below) and its English reads well. I think it is a relevant addition to the literature because (1) it confirms previous results on Chinese coal mine methane emissions with an original method; and (2) it builds more confidence and understanding of the wind-assigned anomaly method and its sensitivities.

I recommend the publication in ACP once all the following comments are addressed.

**Significant comment on structure, method and data description, and naming**

While concision is indeed a quality when writing a scientific article, the authors must be careful to provide enough information so that it can still be read as a standalone piece. In its current state, this article cuts too many corners in describing their datasets and methods to be read smoothly and requires, on this matter, a significant adjustment.

**Data set descriptions**

The text keeps referring to "the bottom-up inventory" (in the abstract !!, line 20, and at lines 141, 147, 166, 171, 183, 228, 232, 236, 258 and 275) which is different from commonly-used EDGAR or CAMS-GLOB-ANT, but without ever properly presenting this different inventory. The reader has only the captions of Figures 2 or 4 to rely on to guess that "the bottom-up inventory" is actually work by Qin et al. (2023). Considering that the Qin et al. (2023) bottom-up inventory is a significant discussion reference, it needs to be clearly presented in the abstract, and presented and described in the main text.

For clarity, I would suggest to gather the descriptions of all three emission inventories (Qin et al. (2023), EDGAR and CAMS-GLOB-ANT) in a dedicated subsection of Section 2 "Data and method".

In addition, regarding naming, the expression "the bottom-up inventory" which is repeatedly used to refer to Qin et al. (2023) may be confusing to some readers as EDGAR and CAMS-GLOB-ANT can also be understood and referred-to as bottom-up inventories (e.g. Janssens-Maenhout et al., 2019). I would suggest to use the actual citation or a defined abbreviation/acronym to refer to the Qin et al. (2023) bottom-up inventory in the text, and in Figures captions and labels.

**Method description**

Subsection 2.2 named "Wind-assigned anomaly method", only briefly provides the Tu et al. (2022a) reference that actually gives the description of the wind-assigned anomaly method, and then just details the approximate "cone plume" model.

These few elements are insufficient for a standalone reading and understanding of the work performed in this study. While it is unnecessary to reproduce all the description and equations provided in the main text and appendices by Tu et al. (2022a), a 1-2 paragraph digest description of how the method works is at least expected.

The reading would be greatly improved if such a 1-2 paragraph digest description of the wind-assigned anomaly would mention and provide the minimally-required information on: (1) the background estimation and removal; (2) the principles of averaging TROPOMI data for two different wind field segmentations and making the difference of those averages; (3) the fitting of modelled against observed wind-assigned anomalies to estimate an emission scaling factor; and (4) uncertainty bar calculation.

Besides, the discussion of using a Gaussian plume instead of a cone plume model included in this work shows that the cone plume model, in itself, is not essential to the wind-assigned anomaly method. For clarity, I would thus suggest to separate the descriptions of the wind-assigned anomaly method principles and approximate plume models in different sub(-sub?)sections, and so also move the description of the Gaussian plume model from Sect.3 to somewhere in Sect 2.

**Significant comments on Results and discussion**

The Results and discussion section can be improved on three different aspects, detailed below.

*Discussion on elevation features, approximate plume models and their opening-angle parameters*

To my understanding, this application of the wind-assigned anomaly method in Shanxi brings a new additional interesting aspect that is not currently discussed in the paper. Works by Tu et al. (2022a,b) previously studied locations where methane can be transported in plumes over relatively flat terrains, along elevation features. However, this new study area in Shanxi has elevation features all around the target sources, methane is being blown against these elevation features and piles up at the bottom of valleys sources.

I would expect that approximate plume models struggle more to reproduce realistic enhancements in mountainous areas with complex elevation features such as Shanxi, compared to flatter terrains like the ones near Madrid or around the Polish coal mines. Interestingly, the discussion in Sect 3.4.1 shows for Changzhi region that changing the approximate plume model from cone to Gaussian improves the wind-assigned anomaly result comparison to Qin et al. (2023). Furthermore, it also shows that increasing the opening angle of these approximation models from 60° upwards improves the comparison even more.

Does that also hold for Jincheng and Yangquan regions? Could the Gaussian plume model with wider opening angles, to some extent, be more appropriate to approximate transport in such mountainous areas with complex elevation features compared to the cone model with the lower opening angle of 60°? What could be explored to test such an hypothesis? Could test

experiments with N2O help, like what was done near Madrid in Tu et al. (2022a) to show that wind-assigned anomaly works?

I do not obviously expect that all these questions will be precisely answered after completing the review process. However, I think that additional discussion elements on the appropriateness of different approximate plume models and/or of their opening angle parameter values, possibly in relation with complex elevation features in Shanxi, can be an interesting and valuable addition.

*Revising uncertainty estimates to account for method-related errors*

Currently, the uncertainty estimates, which seem to correspond to the error bars in Figure 6, amount to only a few percents of the total emission estimates: 1.9%, 1.4% and 3.7% for Changzhi, Jincheng and Yangquan regions, respectively. From Sect. 2 (see significant comment on method description above), I can only guess that these uncertainty estimates include the contributions of background estimation error and satellite data noise, as performed in Tu et al. (2022a,b).

In addition, Sect. 3.4 discusses the impact of (1) changing the approximate plume model to the Gaussian plume model, and perturbing the opening angle; (2) changing the wind product from ERA5 to NCEP, and changing the direction segmentation; and (3) changing the a priori inventories. Besides, other parameters may influence the results as well, such as the height of the wind speed, as discussed in Tu et al. (2022b): why is 100m chosen in this work, whereas 10m was used for Madrid area in Tu et al. (2022a), and 330m for Poland, in Tu et al. (2022b)?

Overall, the sensitivity tests reported here result in emission rates changing from -5% up to +12%, which is larger in magnitude than the maximum of the currently reported uncertainties. As the choice of many method inputs and parameters can be somewhat arbitrary (ERA5 winds against NCEP, Gaussian against cone plume, 60° against 80° fov, wind speed height, etc), the uncertainty estimates provided in this work need to be revised to account for the contribution of these method-related choices and uncertainties.

For example, ways to account for these method-related uncertainties can be through the definition of a comprehensive quantification ensemble, which explores reasonable ranges for different method inputs and parameter values, such as done by e.g. Schuit et al. (2023), or to sum different contributions in quadrature, such as done by e.g. Cusworth et al. (2021).

Revised uncertainties are expected to be higher. However, these larger uncertainties may actually help to better compare with emission rates reported by Qin et al. (2023), and to assess how significant the difference found between EDGAR/CAMS-GLOB-ANT and wind-assigned anomaly results is.

*Discussion of results against previous satellite-based top-down estimates*

Satellite-based estimates of methane emissions are currently being studied and developed at different scales, with different datasets and different methods, by several groups across the world. For example, Chen et al. (2022) report a downward correction of coal mine emissions

in China compared to UNFCCC reports, partly driven by Shanxi; or Zhang et al. (2021) report a 30% decrease in their posterior estimates for China, 60% of which are attributed to coal mines, and provide an extensive list of other studies supporting a consistent result.

These previous studies and/or others need to be mentioned in this article. They could for example be included in the Introduction, and their relevant messages cited and discussed when presenting the wind-assigned anomaly results. As they give a similar picture of overestimated Shanxi coal mine methane emissions in EDGAR/CAMS-GLOB-ANT, the overall article message would be even more highlighted, while at the same time building more confidence in the wind-assigned anomaly method.

**Minor corrections and questions**

- Section 2.1 "TROPOMI dataset": Please provide the data quality filters applied to select the TROPOMI data included in this work.

- Line 148: Please delete the "latest" adjective for EDGAR v7, as EDGAR v8 has just been released (https://edgar.jrc.ec.europa.eu/dataset_ghg80).

- Figures 4, 6 A-3 and A-5: Please use a consistent label to designate emissions from Qin et al. (2023), "bottom-up inventory" on one side, and "shaft emission" on the other.

- Figure 7: Please start the y-axis range to 0 in order to facilitate comparison with Figure 6.

- Sect 3.4.3: It is unclear whether the "calculated average" (line 231) refers to simulated or observed averaged enhancement, please precise. If it is simulated, the fact that enhancements are lower with CAMS-GLOB-ANT whereas this inventory prescribes nearly twice as high emissions is quite counter-intuitive and surprising. Is this explained by the sentence lines 234-235 about similar background estimation errors that compensate in the wind-assigned difference? If so, could you please reformulate this explanation and move it a few lines earlier in the text, when "calculated average" values are compared?

- Line 236: I think there is a typo, isn't $8.5 \times 10^{27}$ supposed to be $8.5 \times 10^{26}$ instead? Otherwise, it would mean an order of magnitude difference…

- Figure A-10: Are marker supposed to be missing in the scatter plot (right panel). If not, can you please add them, otherwise explain why there is no marker in the scatter plot?

**References**

Qin, K., Hu, W., He, Q., Lu, F., and Cohen, J. B.: Individual Coal Mine Methane Emissions Constrained by Eddy-Covariance Measurements: Low Bias and Missing Sources, EGUsphere, 1–49, https://doi.org/10.5194/egusphere-2023-1210, 2023.

Janssens-Maenhout, G., Crippa, M., Guizzardi, D., Muntean, M., Schaaf, E., Dentener, F., Bergamaschi, P., Pagliari, V., Olivier, J. G. J., Peters, J. A. H. W., van Aardenne, J. A., Monni, S., Doering, U., Petrescu, A. M. R., Solazzo, E., and Oreggioni, G. D.: EDGAR v4.3.2 Global Atlas of the three major greenhouse gas emissions for the period 1970–2012, Earth Syst. Sci. Data, 11, 959–1002, https://doi.org/10.5194/essd-11-959-2019, 2019.

Tu, Q., Hase, F., Schneider, M., García, O., Blumenstock, T., Borsdorff, T., Frey, M., Khosrawi, F., Lorente, A., Alberti, C., Bustos, J. J., Butz, A., Carreño, V., Cuevas, E., Curcoll, R., Diekmann, C. J., Dubravica, D., Ertl, B., Estruch, C., León-Luis, S. F., Marrero, C., Morgui, J.-A., Ramos, R., Scharun, C., Schneider, C., Sepúlveda, E., Toledano, C., and Torres, C.: Quantification of $CH_4$ emissions from waste disposal sites near the city of Madrid using ground- and space-based observations of COCCON, TROPOMI and IASI, Atmos. Chem. Phys., 22, 295–317, https://doi.org/10.5194/acp-22-295-2022, 2022a.

Tu, Q., Schneider, M., Hase, F., Khosrawi, F., Ertl, B., Necki, J., Dubravica, D., Diekmann, C. J., Blumenstock, T., and Fang, D.: Quantifying $CH_4$ emissions in hard coal mines from TROPOMI and IASI observations using the wind-assigned anomaly method, Atmos. Chem. Phys., 22, 9747–9765, https://doi.org/10.5194/acp-22-9747-2022, 2022b

Schuit, B. J., Maasakkers, J. D., Bijl, P., Mahapatra, G., van den Berg, A.-W., Pandey, S., Lorente, A., Borsdorff, T., Houweling, S., Varon, D. J., McKeever, J., Jervis, D., Girard, M., Irakulis-Loitxate, I., Gorroño, J., Guanter, L., Cusworth, D. H., and Aben, I.: Automated detection and monitoring of methane super-emitters using satellite data, Atmos. Chem. Phys., 23, 9071–9098, https://doi.org/10.5194/acp-23-9071-2023, 2023.

Daniel H. Cusworth, Riley M. Duren, Andrew K. Thorpe, Winston Olson-Duvall, Joseph Heckler, John W. Chapman, Michael L. Eastwood, Mark C. Helmlinger, Robert O. Green, Gregory P. Asner, Philip E. Dennison, and Charles E. Miller, Intermittency of Large Methane Emitters in the Permian Basin, Environmental Science & Technology Letters, DOI: 10.1021/acs.estlett.1c00173, 2021

Chen, Z., Jacob, D. J., Nesser, H., Sulprizio, M. P., Lorente, A., Varon, D. J., Lu, X., Shen, L., Qu, Z., Penn, E., and Yu, X.: Methane emissions from China: a high-resolution inversion of TROPOMI satellite observations, Atmos. Chem. Phys., 22, 10809–10826, https://doi.org/10.5194/acp-22-10809-2022, 2022.

Zhang, Y., Jacob, D. J., Lu, X., Maasakkers, J. D., Scarpelli, T. R., Sheng, J.-X., Shen, L., Qu, Z., Sulprizio, M. P., Chang, J., Bloom, A. A., Ma, S., Worden, J., Parker, R. J., and Boesch, H.: Attribution of the accelerating increase in atmospheric methane during 2010–2018 by inverse analysis of GOSAT observations, Atmos. Chem. Phys., 21, 3643–3666, https://doi.org/10.5194/acp-21-3643-2021, 2021.

---

## Author Comment (AC1)

**Response to Referee #2**

We would like to thank reviewer #2 for taking the time to review this manuscript and for providing valuable, constructive feedback and corresponding suggestions that helped us to further improve the manuscript.

In this author's comment, all the points raised by the reviewer are copied here one by one and shown in blue color, along with the corresponding reply from the authors in black.

This article provides coal mine methane emission estimates for three regions of the Shanxi province (China), using the recently-developed wind-assigned anomaly method with methane concentration observations from the TROPOMI satellite instrument. The results suggest that commonly-used emission inventories overestimate coal mine emissions in the area. The sensitivities of wind-assigned anomaly results to several of the method inputs and parameters are described.

The article is concise (too much actually, see below) and its English reads well. I think it is a relevant addition to the literature because (1) it confirms previous results on Chinese coal mine methane emissions with an original method; and (2) it builds more confidence and understanding of the wind-assigned anomaly method and its sensitivities.

I recommend the publication in ACP once all the following comments are addressed.

We thank the reviewer for this positive statement.

**Significant comment on structure, method and data description, and naming**

While concision is indeed a quality when writing a scientific article, the authors must be careful to provide enough information so that it can still be read as a standalone piece. In its current state, this article cuts too many corners in describing their datasets and methods to be read smoothly and requires, on this matter, a significant adjustment.

We have added more information in the manuscript as suggested by both referees.

*Data set descriptions*

The text keeps referring to "the bottom-up inventory" (in the abstract !!, line 20, and at lines 141, 147, 166, 171, 183, 228, 232, 236, 258 and 275) which is different from commonly-used EDGAR or CAMS-GLOB-ANT, but without ever properly presenting this different inventory. The reader has only the captions of Figures 2 or 4 to rely on to guess that "the bottom-up inventory" is actually work by Qin et al. (2023). Considering that the Qin et al. (2023) bottom-up inventory is a significant discussion reference, it needs to be clearly presented in the abstract, and presented and described in the main text.

For clarity, I would suggest to gather the descriptions of all three emission inventories (Qin et al. (2023), EDGAR and CAMS-GLOB-ANT) in a dedicated subsection of Section 2 "Data and method".

In addition, regarding naming, the expression "the bottom-up inventory" which is repeatedly used to refer to Qin et al. (2023) may be confusing to some readers as EDGAR and CAMS-GLOB-ANT can also be understood and referred-to as bottom-up inventories (e.g. Janssens- Maenhout et al., 2019). I would suggest to use the actual citation or a defined abbreviation/ acronym to refer to the Qin et al. (2023) bottom-up inventory in the text, and in Figures captions and labels.

Thanks to the referee for this suggestion. We have incorporated additional details about the "bottom-up inventory" in the abstract. Furthermore, we have introduced an additional subsection in the "Data and method" section to comprehensively describe all three emission inventories as suggested by the referee. In order to differentiate the presently mentioned "bottom-up inventory" from CAMS-GLOB-ANT and EDGAR, we have revised the name to "IPCC Tier 2 bottom-up inventory (Qin et al., 2023)". The figures are updated accordingly.

*Method description*

Subsection 2.2 named "Wind-assigned anomaly method", only briefly provides the Tu et al. (2022a) reference that actually gives the description of the wind-assigned anomaly method, and then just details the approximate "cone plume" model.

These few elements are insufficient for a standalone reading and understanding of the work performed in this study. While it is unnecessary to reproduce all the description and equations provided in the main text and appendices by Tu et al. (2022a), a 1-2 paragraph digest description of how the method works is at least expected.

The reading would be greatly improved if such a 1-2 paragraph digest description of the wind-assigned anomaly would mention and provide the minimally-required information on: (1) the background estimation and removal; (2) the principles of averaging TROPOMI data for two different wind field segmentations and making the difference of those averages; (3) the fitting of modelled against observed wind-assigned anomalies to estimate an emission scaling factor; and (4) uncertainty bar calculation.

We have added all information related with the wind-assigned method in Subsection 2.3-Dispersion model and 2.4-Background removal and wind-assigned anomaly method, as suggested by the referee.

Besides, the discussion of using a Gaussian plume instead of a cone plume model included in this work shows that the cone plume model, in itself, is not essential to the wind-assigned anomaly method. For clarity, I would thus suggest to separate the descriptions of the wind-assigned anomaly method principles and approximate plume models in different sub(- sub?)sections, and so also move the description of the Gaussian plume model from Sect.3 to somewhere in Sect 2.

Two separated subsections about plume models (cone plume and Gaussian plume) and the wind-assigned anomaly method have been added in Sect. 2.3 and 2.4, as suggested by the referee.

**Significant comments on Results and discussion**

The Results and discussion section can be improved on three different aspects, detailed below.

*Discussion on elevation features, approximate plume models and their opening-angle parameters*

To my understanding, this application of the wind-assigned anomaly method in Shanxi brings a new additional interesting aspect that is not currently discussed in the paper. Works by Tu et al. (2022a,b) previously studied locations where methane can be transported in plumes, along elevation features. However, this new study area in Shanxi has elevation features all around the target sources, methane is being blown against these elevation features and piles up at the bottom of valleys sources.

I would expect that approximate plume models struggle more to reproduce realistic enhancements in mountainous areas with complex elevation features such as Shanxi, compared to flatter terrains like the ones near Madrid or around the Polish coal mines. Interestingly, the discussion in Sect 3.4.1 shows for Changzhi region that changing the approximate plume model from cone to Gaussian improves the wind-assigned anomaly result comparison to Qin et al. (2023). Furthermore, it also shows that increasing the opening angle of these approximation models from 60° upwards improves the comparison even more.

Does that also hold for Jincheng and Yangquan regions? Could the Gaussian plume model with wider opening angles, to some extent, be more appropriate to approximate transport in such mountainous areas with complex elevation features compared to the cone model with the lower opening angle of 60°? What could be explored to test such an hypothesis? Could test experiments with N2O help, like what was done near Madrid in Tu et al. (2022a) to show that wind-assigned anomaly works?

I do not obviously expect that all these questions will be precisely answered after completing the review process. However, I think that additional discussion elements on the appropriateness of different approximate plume models and/or of their opening angle parameter values, possibly in relation with complex elevation features in Shanxi, can be an interesting and valuable addition.

Thanks to the referee for suggesting a valuable approach to identify optimal opening angles. We have tested experiments with TROPOMI $NO_2$ in the study regions. However, the complexity of the spatial distribution of $NO_2$ sources in these regions became apparent. The $NO_2$ sources are in the Changzhi region (see Figure 1-left) differ significantly from those in Madrid, where the dominant sources are concentrated in the city center. Notably, the wind-assigned anomalies (i.e., the difference in the TROPOMI tropospheric $NO_2$ concentration under NW and SE wind regimes) does not exhibit a distinct bipolar plume (see Figure 1-right). Thus, the $NO_2$ test experiment unfortunately proves ineffective in the study area.

[Figure]

Figure 1: (left) spatial distribution of TROPOMI tropospheric $NO_2$, (right) wind-assigned anomalies (NW − SE) based on TROPOMI tropospheric $NO_2$ in Changzhi region.

Elevated concentrations of $XCH_4$ are notably centered in the heart of the Changzhi region, with slightly lower values observed in the southern areas, where coal mines are clustered (Figure 2 left). Taking into consideration the elevation features, the central part of the Changzhi region is characterized by flat terrain, while elevations rise in the northeast and southeast, as depicted in Figure 2 on the right. Thus, we expect that $CH_4$ does not accumulate at the bottom of valleys but tends to distribute across the entire flat terrain. The orientation of the mountains supports certain prevailing wind patterns in the study area, which is reflected by our segmentation choice.

[Figure]

Figure 2: spatial distribution of XCH₄ (left) and altitude (right) from TROPOMI observations.

The impact of two distinct dispersion models (Gaussian plume and cone plume) on the estimated emission rates in distinct study regions is depicted in Figure 3. Notably, estimates show an upward trend with higher opening angle (fov) values for both models. Furthermore, the difference in estimates between the two models become more pronounced as fov values increase. The estimated emission occurs closest to the bottom-up inventory for higher fov in Changzhi region (i.e., 80º), while a closer match is observed for lower fov values in the Jincheng and Yangquan regions (i.e., 20º).

It appears difficult to decide whether using wider opening angles for approximating in such mountainous areas is the superior choice. Instead, it is more appropriate to view the opening angle as a contributing factor to uncertainty in estimating emission rates.

[Figure]

Figure 3: Estimates of emission rates in Changzhi, Jincheng and Yangquan regions with respect to different opening angles based on cone plume and Gaussian plume. The three different inventories are presented as well.

To further investigate the wind pattern on the complex terrain, we subdivide the study area in the Changzhi region into three subregions (specifically, areas between 35.8 º – 36.3º N, 36.3 º – 36.8º N, 36.8 º – 37.2º N). The corresponding wind rose plots are illustrated in Figure 4. The wind patterns in the southern and central subregions, where most coal mines are located, exhibit a similar pattern, whereas the northern subregion tends to feature more wind from the NW direction and less from the SE direction. Thus, the wind distribution appears generally homogenous in areas with dominant emission sources, while the complex terrain demonstrates a more pronounced impact on the northern region where less coal mines are located.

[Figure]

Figure 4: Wind roses plots for local daytime (08:00–18:00 UTC) from May 2018 to April 2023 for the ERA5 model wind for three subregions in Changzhi. The region range is given in the title of each subfigure.

To further investigate the sensitivity of the wind spatial variation, the wind data at the central point (36.5º N, 113º E for Changzhi, 35.5º N, 112.75º E for Jincheng and 38º N, 113.5º E in Yangquan) is used as a representative value to represent the wind for the entire study area. This substitution results in a decreased estimated emission rate of 11% (emission rate: $7.5 \times 10^{26}$ molec. s$^{-1}$) in Changzhi, an increased emission rate of 7% ($1.5 \times 10^{27}$ molec. s$^{-1}$) in Jincheng and of 8% ($5.3 \times 10^{26}$ molec. s$^{-1}$) in Yangquan. The discussion of this uncertainty has been included in Subsection 3.3.3 in the revised manuscript.

*Revising uncertainty estimates to account for method-related errors*

Currently, the uncertainty estimates, which seem to correspond to the error bars in Figure 6, amount to only a few percents of the total emission estimates: 1.9%, 1.4% and 3.7% for Changzhi, Jincheng and Yangquan regions, respectively. From Sect. 2 (see significant comment on method description above), I can only guess that these uncertainty estimates include the contributions of background estimation error and satellite data noise, as performed in Tu et al. (2022a,b).

Thanks to the referee for emphasizing the uncertainty issues. The current uncertainty, as mentioned by the referee, includes only the contributions of background estimation error and satellite data noise. In response to this concern, we have expanded the discussion on additional sources of uncertainty in the revised manuscript (see section 3.3) and have accordingly update the total uncertainties.

In addition, Sect. 3.4 discusses the impact of (1) changing the approximate plume model to the Gaussian plume model, and perturbing the opening angle; (2) changing the wind product from ERA5 to NCEP, and changing the direction segmentation; and (3) changing the a priori inventories. Besides, other parameters may influence the results as well, such as the height of the wind speed, as discussed in Tu et al. (2022b): why is 100m chosen in this work, whereas 10m was used for Madrid area in Tu et al. (2022a), and 330m for Poland, in Tu et al. (2022b)?

Concerning wind at different height levels, we employed wind data at 10 m for the Madrid area due to the availability of meteorological station data measuring wind at this height. The in-situ measured wind at 10 m was employed together with ground-based measurements (EM27/SUN) to estimate local emission rates.

In the Poland study, we firstly used $XCH_4$ from the CAMS and its corresponding emissions to assess the wind-assigned anomaly method. Because the study area was larger (so we expect more vertical mixing), we assumed that using winds at a higher level of 330 m would provide a superior description of the transport on larger scales. Sensitivity analyses investigating the uncertainty connected to the altitude choice were conducted in both studies.

Overall, the sensitivity tests reported here result in emission rates changing from -5% up to +12%, which is larger in magnitude than the maximum of the currently reported uncertainties. As the choice of many method inputs and parameters can be somewhat arbitrary (ERA5 winds against NCEP, Gaussian against cone plume, 60° against 80° fov, wind speed height, etc), the uncertainty estimates provided in this work need to be revised to account for the contribution of these method-related choices and uncertainties.

For example, ways to account for these method-related uncertainties can be through the definition of a comprehensive quantification ensemble, which explores reasonable ranges for different method inputs and parameter values, such as done by e.g. Schuit et al. (2023), or to sum different contributions in quadrature, such as done by e.g. Cusworth et al. (2021).

Revised uncertainties are expected to be higher. However, these larger uncertainties may actually help to better compare with emission rates reported by Qin et al. (2023), and to assess how significant the difference found between EDGAR/CAMS-GLOB-ANT and wind-assigned anomaly results is.

Thanks to the referee for this important comment. The model and the input data represent the primary contributors to uncertainties. To encompass a comprehensive understanding of these uncertainties, we divided our analysis into four key components in the revised manuscript: 1) background removal (10[th] lower percentile of overall satellite observations each day as the background); 2) dispersion model, including Gaussian against cone plume and variations in fov; 3) wind information, covering wind at different heights, wind data from different sources, wind segmentation and spatial variations; and 4) inventory, serving as the a priori knowledge (CAMS-GLOB-ANT inventory against the IPCC Tier 2 bottom-up inventory (Qin et al., 2023)).

The total uncertainty is computed through error propagation, similar to the approach of summing contributions in quadrature as done by Cusworth et al. (2021). This computation yields a total uncertainty of 25% in Changzhi, 20% in Jincheng and 21% in Yangquan.

*Discussion of results against previous satellite-based top-down estimates*

Satellite-based estimates of methane emissions are currently being studied and developed at different scales, with different datasets and different methods, by several groups across the world. For example, Chen et al. (2022) report a downward correction of coal mine emissions in China compared to UNFCCC reports, partly driven by Shanxi; or Zhang et al. (2021) report a 30% decrease in their posterior estimates for China, 60% of which are attributed to coal mines, and provide an extensive list of other studies supporting a consistent result.

These previous studies and/or others need to be mentioned in this article. They could for example be included in the Introduction, and their relevant messages cited and discussed when presenting the wind-assigned anomaly results. As they give a similar picture of overestimated Shanxi coal mine methane emissions in EDGAR/CAMS-GLOB-ANT, the overall article message would be even more highlighted, while at the

same time building more confidence in the wind-assigned anomaly method.

Thanks to the referee for bringing this valuable information to our attention. We have added related information in the introduction and the results and discussion part as recommended by the referee.

In introduction:
"Qu et al. (2021) highlighted significant challenges in their satellite inversion over southeast China characterized by elevated seasonal rice emissions that coincide with extensive cloud cover and potential misallocation of coal emission. A recent study from Chen et al. (2022) reported a downward correction in CMM emissions (-15%) in China compared to the United Nations Framework Convention on Climate Change (UNFCCC) reports, partly driven by Shanxi. Zhang et al. (2021) documented an overestimation of anthropogenic emissions from China, revealing a 30% decrease in the posterior estimates, with approximately 60% of this downward correction attributed to coal mining."

In Results and discussion:
"Our CMM estimates in these three regions fall within the 30th and 70th percentile range of the updated emission rates in the study by Qin et al. (2023). In addition, our results are consistently lower than the CAMS-GLOB-ANT and the EDGARv7 inventories. This result agrees with previous studies. For instance, a -15% underestimation compared to the UNFCCC has been reported by Chen et al. (2022). Additionally, Zhang et al. (2021) documented a 30% decrease in their posterior estimates for China, with 60% attributed to coal mining. This pattern of overestimation in anthropogenic emissions, in comparison to China's inventory, has been observed in previous research, utilizing GOSAT inversion and various versions of the EDGAR inventory as a priori estimates (Miller et al., 2019; Maasakkers et al., 2019). This divergence may be attributed to two reasons: (1) missing observation of strong CMM emissions during the TROPOMI overpass. It is important to note that CMM emissions exhibit a strong dependency on coal mine activities, which vary over time. The TROPOMI data provide instantaneous observations, capturing $CH_4$ concentrations at a specific moment (local time ~ 13:30), thereby leading to limitations in detecting strong CMM emissions during both morning and afternoon periods. (2) the CMM utilization have been largely improved in the last decade, since the national government issued specific targets in the national 12th and 13th five-year plan (Gao et al., 2021; Lu et al., 2021)."

**Minor corrections and questions**

- Section 2.1 "TROPOMI dataset": Please provide the data quality filters applied to select the TROPOMI data included in this work.

Thank you. The information has been added.
"A data quality filter (qa = 1.0) is applied to characterize the data during clear-sky and low-cloud atmospheric conditions."

- Line 148: Please delete the "latest" adjective for EDGAR v7, as EDGAR v8 has just been released (https://edgar.jrc.ec.europa.eu/dataset_ghg80).

Thank you for providing this information. Corrected.

- Figures 4, 6 A-3 and A-5: Please use a consistent label to designate emissions from Qin et al. (2023), "bottom-up inventory" on one side, and "shaft emission" on the other.

Thank you. Corrected.

- Figure 7: Please start the y-axis range to 0 in order to facilitate comparison with Figure6.

Thank you. The figure has been updated.

- Sect 3.4.3: It is unclear whether the "calculated average" (line 231) refers to simulated or observed averaged enhancement, please precise. If it is simulated, the fact that enhancements are lower with CAMS-GLOB-ANT whereas this inventory prescribes nearly twice as high emissions is quite counter-intuitive and surprising. Is this explained by the sentence lines 234-235 about similar background estimation errors that compensate in the wind-assigned difference? If so, could you please reformulate this explanation and move it a few lines earlier in the text, when "calculated average" values are compared?

The "calculated average" refers to the observed enhancement (i.e., TROPOMI $XCH_4$ – background). This information has been detailed in the text. The a-priori inventory information, including the location and emission rates of the sources, has a small impact on the estimation of the background, as illustrated in the left figure below. The difference ($1.12 \pm 2.93$ ppb, $R^2 = 0.8562$) arising from the use of different inventories as the a priori, is effectively mitigated ($R^2 = 0.9962$) when comparing the wind-assigned anomalies, as illustrated in the right figure below. It is because the systematic errors in background removal is compensated by computing the differences of enhancements under different wind field segmentations. These figures are presented in Figure A- 14 in the updated manuscript.

[Figure]

- Line 236: I think there is a typo, isn't 8.5 x 10^27 supposed to be 8.5 x 10^26 instead? Otherwise, it would mean an order of magnitude difference…

Thank you. Corrected.

- Figure A-10: Are marker supposed to be missing in the scatter plot (right panel). If not, can you please add them, otherwise explain why there is no marker in the scatter plot?

We appreciated the referee for bringing this to our attention. In addition to the missing marker, we have also recognized that the current correlation figure, similar to the Figure 5(c) (or Figure A-6) in the manuscript, might raise confusion for readers.

The existing right figure differs from Figure 5(c) in the manuscript, which illustrates anomalies derived from the cone plume model and TROPOMI observations. It presents the correlation of modelled wind-assigned anomalies using different inventories, i.e., the bottom-up inventory from Qin et al. (2023) and the CAMS-GLOB-ANT inventory. It aims to display how different inventories affect anomalies based on the cone plume model. To avoid potential confusion for readers, the correlation plot, featuring different inventories, is presented in a distinct plotting style and as a subfigure in Figure A-14.

[Figure]

**References**

Qin, K., Hu, W., He, Q., Lu, F., and Cohen, J. B.: Individual Coal Mine Methane Emissions Constrained by Eddy-Covariance Measurements: Low Bias and Missing Sources, EGUsphere, 1–49, https://doi.org/10.5194/egusphere-2023-1210, 2023.

Janssens-Maenhout, G., Crippa, M., Guizzardi, D., Muntean, M., Schaaf, E., Dentener, F., Bergamaschi, P., Pagliari, V., Olivier, J. G. J., Peters, J. A. H. W., van Aardenne, J. A., Monni, S., Doering, U., Petrescu, A. M. R., Solazzo, E., and Oreggioni, G. D.: EDGAR v4.3.2 Global Atlas of the three major greenhouse gas emissions for the period 1970–2012, Earth Syst. Sci. Data, 11, 959–1002, https://doi.org/10.5194/essd-11-959-2019, 2019.

Tu, Q., Hase, F., Schneider, M., García, O., Blumenstock, T., Borsdorff, T., Frey, M., Khosrawi, F., Lorente, A., Alberti, C., Bustos, J. J., Butz, A., Carreño, V., Cuevas, E., Curcoll, R., Diekmann, C. J., Dubravica, D., Ertl, B., Estruch, C., León-Luis, S. F., Marrero, C., Morgui, J.-A., Ramos, R., Scharun, C., Schneider, C., Sepúlveda, E., Toledano, C., and Torres, C.: Quantification of CH4 emissions from waste disposal sites near the city of Madrid using ground- and space-based observations of COCCON, TROPOMI and IASI, Atmos. Chem. Phys., 22, 295–317, https://doi.org/10.5194/acp-22-295-2022, 2022a.

Tu, Q., Schneider, M., Hase, F., Khosrawi, F., Ertl, B., Necki, J., Dubravica, D., Diekmann, C. J., Blumenstock, T., and Fang, D.: Quantifying CH4 emissions in hard coal mines from TROPOMI and IASI

observations using the wind-assigned anomaly method, Atmos. Chem. Phys., 22, 9747–9765, https://doi.org/10.5194/acp-22-9747-2022, 2022b

Schuit, B. J., Maasakkers, J. D., Bijl, P., Mahapatra, G., van den Berg, A.-W., Pandey, S., Lorente, A., Borsdorff, T., Houweling, S., Varon, D. J., McKeever, J., Jervis, D., Girard, M., Irakulis-Loitxate, I., Gorroño, J., Guanter, L., Cusworth, D. H., and Aben, I.: Automated detection and monitoring of methane super-emitters using satellite data, Atmos. Chem. Phys., 23, 9071–9098, https://doi.org/10.5194/acp-23-9071-2023, 2023.

Daniel H. Cusworth, Riley M. Duren, Andrew K. Thorpe, Winston Olson-Duvall, Joseph Heckler, John W. Chapman, Michael L. Eastwood, Mark C. Helmlinger, Robert O. Green, Gregory P. Asner, Philip E. Dennison, and Charles E. Miller, Intermittency of Large Methane Emitters in the Permian Basin, Environmental Science & Technology Letters, DOI: 10.1021/acs.estlett.1c00173, 2021

Chen, Z., Jacob, D. J., Nesser, H., Sulprizio, M. P., Lorente, A., Varon, D. J., Lu, X., Shen, L., Qu, Z., Penn, E., and Yu, X.: Methane emissions from China: a high-resolution inversion of TROPOMI satellite observations, Atmos. Chem. Phys., 22, 10809–10826, https://doi.org/10.5194/acp-22-10809-2022, 2022.

Zhang, Y., Jacob, D. J., Lu, X., Maasakkers, J. D., Scarpelli, T. R., Sheng, J.-X., Shen, L., Qu, Z.,

Sulprizio, M. P., Chang, J., Bloom, A. A., Ma, S., Worden, J., Parker, R. J., and Boesch, H.: Attribution of the accelerating increase in atmospheric methane during 2010–2018 by inverse analysis of GOSAT observations, Atmos. Chem. Phys., 21, 3643–3666, https://doi.org/10.5194/acp-21-3643-2021, 2021.

---

## Author Comment (AC2)

**Response to Referee #1**

We would like to thank reviewer #1 for taking the time to review this manuscript and for providing valuable, constructive feedback and corresponding suggestions that helped us to further improve the manuscript.

In this author's comment, all the points raised by the reviewer are copied here one by one and shown in blue color, along with the corresponding reply from the authors in black.

Tu et al. present an analysis of TROPOMI methane observations over the coal-rich Shanxi province of China. They use their wind-assigned anomaly method to quantify regional methane emissions for three clusters of Shanxi coal mines. They compare their estimates with three bottom-up emission inventories, EDGAR v7.0, CAMS-GLOB-ANT, and a measurement-based coal mine methane inventory by Qin et al. (2023). They find good agreement with the Qin et al. estimates but much lower emissions (~factor of 2-3) than reported in the EDGAR and CAMS inventories.

The paper is interesting and a good fit for ACP, but in my view substantial changes are needed before it can be published. I see two major weaknesses. First, the methods need to be explained in much more detail, not merely by pointing to previous publications. I found it very difficult to follow the discussion of results because the paper does not adequately explain the wind-assigned anomaly method and its interpretation. Second, the uncertainty analysis appears to be incomplete. The authors report uncertainties <4% (<2% in two of three cases) on their regional methane emission estimates inferred from TROPOMI. These values are unrealistically low; regional emission errors reported elsewhere in the literature are routinely in the range ~20%-30%. I am therefore left with the impression that the authors have overlooked important sources of error, for example having to do with background subtraction and wind speed.

We appreciate that the referee provided us these valuable comments. With the assistance of these comments, we have tried to improve the manuscript accordingly.

More information has been added in the manuscript as suggested by both referees. The dispersion model and the wind-assigned anomaly method have been explained in detail in Section 2. To address concerns related to uncertainty analysis, we have not only considered background estimation error and satellite data noise, but have also discussed the uncertainties associated with the dispersion model and its inputs. The new derived errors are in the range of 20%-25%. These two issues will be discussed in detail below.

**Specific comments**

- L. 18-19: The reported uncertainties (<4%) are unrealistically small. Uncertainty in regional emissions derived from TROPOMI tend to be in the 20%-30% range (or more). There must be other, larger sources of error besides what is reported.

- Shen et al. (2022) used TROPOMI to estimate methane emissions for ~20 US oil and gas basins and reported mean errors of 30% based on an elaborate uncertainty analysis (https://acp.copernicus.org/articles/22/11203/2022/). Error bars for emissions from individual countries estimated by Shen et al. (2023) are of similar magnitude

(https://www.nature.com/articles/s41467-023-40671-6). TROPOMI analyses by Cusworth et al. (2022), Chen et al. (2023), and many others found similar results.

The previously reported uncertainties included only the contributions from background estimation error and satellite data noise. In the revised version of the manuscript, we also discuss the uncertainties arising from the background removal, dispersion model (cone plume model or Gaussian plume model) and its inputs (wind at different height level, different wind segmentation, the spatial variation of the winds and inventories as the apriori knowledge) and we propagate these uncertainties into our emission estimates. As a result, the total uncertainties of our emission estimates are determined to be 25% in Changzhi, 20% in Jincheng and 21% in Yangquan.

- L. 20: Which bottom-up inventory?

The bottom-up inventory computed based on the IPCC Tier 2 approach from Qin et al. (2023). This information has been included in the abstract and further detailed information is now provided in an additional subsection (Section 2.2).

- L. 21-22: That may be, but it's not entirely clear given the unrealistically small uncertainties reported for the TROPOMI emission estimates.

We have revised the given uncertainties, previously derived only from background estimation error and satellite data noise, to now encompass the specific uncertainties, including those arising from the background removal method, dispersion model and its inputs originated from wind and a priori inventory. The updated uncertainties are computed based on the error propagation, considering all the impacts mentioned above.

- L. 23: How do the estimates help to develop climate mitigation strategies?

The term "develop" might not be a good fit here. We changed this phrase to "provide additional insights (eg. a more realistic approximation based on the measurement dataset) into CMM emissions mitigation".

- L. 31: Would it not be more accurate to say that China is "the leading emitter", rather than just "one of" them?

Methane exhibits a long atmospheric lifetime, showing its influence on a climatic scale rather than an annual one. Meanwhile, certain emission sources, such as those originating from the military, are presently excluded from consideration. Therefore, we suggest the term "one of the leading $CH_4$ emitters" to more accurately convey the significant impact of methane emissions in the broader context of climate implications, given its persistent nature and the exclusion of specific sources.

- L. 33: China did not sign the 2021 Global Methane Pledge, so for clarity it would be best to use another word besides "pledge" here.

The word has been replaced with "committed". China has also signed other agreements and this information has been added in the manuscript.

"China has demonstrated its commitment to addressing $CH_4$ emissions by signing key international agreements such as the Kyoto Protocol in 1998 and the Paris Agreement in 2016, reflecting its dedication to global efforts in mitigating climate change. Additionally, in 2021, China committed to reduce $CH_4$ emissions under the Glasgow Agreement and intended to develop a comprehensive and ambitious National Action Plan with the goal of

achieving a substantial impact on methane emission control and reductions in the 2020s (USDoS, 2021)."

- L. 33-35: It would be useful to include a reference for the Glasgow Agreement. Perhaps something like this 2021 US State Department press release: https://www.state.gov/u-s-china-joint-glasgow-declaration-on-enhancing-climate-action-in-the-2020s/

Thank you. This reference has been added.

- L. 68: It's unclear what "solar radiation […] radiated from the Earth" means.

original sentence: "The instrument utilizes passive remote-sensing techniques to measure solar radiation reflected by and radiated from the Earth across the ultraviolet (UV), visible (VIS), near-infrared (NIR), and short-wave spectral (SWIR) bands (Veefkind et al., 2012)."

the sentence has been changed to "The instrument utilizes passive remote-sensing techniques to measure the backscattered solar radiation across the ultraviolet (UV), visible (VIS), near-infrared (NIR), and short-wave spectral (SWIR) bands (Veefkind et al., 2012)."

- L. 88: What wind speed is used? The speed at 10-m? 50-m? Something else?

Wind data at 100 m is used here. This information has been added and it is also mentioned in Section 3.3. Uncertainty of wind at different height, e.g., 10 m, has been also discussed as part of the error analysis.

- L. 95: Regions of China or of Shanxi?

Thank you. It should be "regions of Shanxi". This has been corrected.

- Figure 1: Suggest increasing font size for legibility.

Thank you. The figure has been updated.

- L. 104-105: What are those estimates by Qin et al. (2023) based on? A brief description of the dataset would be valuable.

Thank you. The description of the dataset has been added in Section 2.2.

"Qin et al. (2023) used both public and private datasets from over 600 individual coal mines in Shanxi Province. The IPCC Tier 2 approach is applied to calculate the corresponding $CH_4$ emissions based on 3-5 sets of observed emission factors, thereby establishing a range of bottom-up estimation of CMM on a mine-by-mine basis. In the following work, the bottom-up inventory computed from the median emission factors (E5) will serve as a prior information in the wind-assigned method for estimating emissions, referring to IPCC Tier 2 bottom-up inventory. In their study, an eddy-covariance tower was installed in Changzhi during two two-month periods to derive an average observed $CH_4$ flux. Based on the in-situ measurements, a series of scaling factors at different percentiles of the observational distribution (i.e., 10%, 30%, 50%, 70%, 90%) were generated. These scaling factors were subsequently employed to update the preliminary Tier 2 bottom-up inventory (Qin et al., 2023). The scaling factors for a specific percentile of the observational distribution show minimal variations among different coal mines, suggesting these factors can be treated as constant values across the ensemble of coal mines at each percentile. Our wind-assigned method emphasizes the proportional share of emissions per mine rather than absolute values, resulting in estimated CMM emissions that do not significantly differ whether using the Tier 2 bottom-up inventory or one of the scaled inventory datasets. In additional to the current

IPCC 2 Tier bottom-up inventory, the scaled inventory is also provided as an additional reference point in this work."

- L. 105-106: I do not understand the sentence beginning "Near 30 small coal mines…"

original sentence: "Near 30 small coal mines scatter in the mountain area in the south and the emissions are relatively small with 24 orders of magnitude in molec. s$^{-1}$".

The coal mines in the south of Zhangzi emit relatively smaller $CH_4$ than the others. There are approximately 30 small coal mines scattered in the mountainous area in the south. The emissions from these mines are relatively small, with values less than $1 \times 10^{25}$ molec. s$^{-1}$ per mine.

The sentence has been rephrased in the manuscript to "There are near 30 small coal mines scattered in the mountain area in the south and each mine has a relatively low emission rate, measuring less than $1.0 \times 10^{25}$ molec. s$^{-1}$."

- L. 110: Qin et al. (2023) used eddy covariance measurements to construct their facility-scale inventory. Would it be appropriate to describe their work as a measurement-based inventory of coal mine emissions?

The referee correctly points out that the final inventory in Qin et al. (2023) uses a mixture of bottom-up and top-down approaches, i.e., using the in-situ measurements to update the preliminary inventory derived from the IPCC Tier 2 approach. From these observations, a series of scaling factors at different percentiles of the observational distribution (i.e., 10%, 30%, 50%, 70%, 90%) were generated. The scaling factors for a specific percentile of the observational distribution show minimal variations among different coal mines, suggesting these factors can be treated as constant values across the ensemble of coal mines at each percentile. Our wind-assigned method emphasizes the proportional share of emissions per mine rather than absolute values, resulting in estimated CMM emissions that do not significantly differ whether using the Tier 2 bottom-up inventory or one of the scaled inventory datasets. In additional to the current IPCC 2 Tier bottom-up inventory, the scaled inventory is also provided as an additional reference point in this work.

We have added detailed information about the bottom-up inventory in Section 2.2. Additionally, to distinguish this bottom-up inventory with other inventories, like the scaled inventory in Qin et al. (2023), CAMS-GLOB-ANT or EDGARv7.0, we use "IPCC Tier 2 bottom-up inventory (Qin et al., 2023).

- Subsection 3.2: Suggest moving this subsection to section 2 (data and methods), including description of the Qin et al. (2023) dataset.

Thank you. We have moved the subsection 3.2 to Section 2 and added more information about the inventory from Qin et al. (2023) in the text as suggested by the referee.

"Qin et al. (2023) used both public and private datasets from over 600 individual coal mines in Shanxi Province. The IPCC Tier 2 approach is applied to calculate the corresponding $CH_4$ emissions based on 3-5 sets of observed emission factors, thereby establishing a range of bottom-up estimation of CMM on a mine-by-mine basis. In the following work, the bottom-up inventory computed from the median emission factors (E5) will serve as a prior information in the wind-assigned method for estimating emissions, referring to IPCC Tier 2 bottom-up inventory. In their study, an eddy-covariance tower was installed in Changzhi during two two-month periods to derive an average observed $CH_4$ flux. Based on the in-situ measurements, a series of scaling factors at different percentiles of the observational

distribution (i.e., 10%, 30%, 50%, 70%, 90%) were generated. These scaling factors were subsequently employed to update the preliminary Tier 2 bottom-up inventory (Qin et al., 2023). The scaling factors for a specific percentile of the observational distribution show minimal variations among different coal mines, suggesting these factors can be treated as constant values across the ensemble of coal mines at each percentile. Our wind-assigned method emphasizes the proportional share of emissions per mine rather than absolute values, resulting in estimated CMM emissions that do not significantly differ whether using the Tier 2 bottom-up inventory or one of the scaled inventory datasets. In additional to the current IPCC 2 Tier bottom-up inventory, the scaled inventory is also provided as an additional reference point in this work."

- Figure 4 (left): A log y-scale would be helpful here.

The figure has been revised, so as the Figure A- 2 for EDGARv7 inventory.

- Section 3.3 and Fig. 5: Significantly more explanation is needed on how the wind-assigned anomalies are calculated and how the wind direction segmentation is performed and what these things mean. The methods section on the wind assigned anomaly method only describes the cone plume model. Section 3.3 is very difficult to follow, and readers shouldn't need to read the authors' previous papers to understand what is going on; the paper should be readable on its own. It's unclear to me what the middle panel of Fig. 5 is showing. How are the estimated emissions distributed between the different coal mines within a cluster? How are the emissions calculated from the TROPOMI wind-assigned anomalies? Are all the mines scaled up/down together following the spatial distribution of the underlying inventory?

Thanks to the referee for this comment. We have added additional details about the dispersion model and the wind-assigned anomaly method in Section 2.

The middle panel of Fig. 5 represents wind-assigned anomalies, representing the difference in TROPOMI enhancements between two wind segmentation. It is important to note that the estimated emission in this context is a total value for the entire study region, rather than a spatial distribution. The inventory, serving as a priori knowledge, provides the CMM emission fractions instead of their absolute values. Thus, as mentioned by the referee, all the mines can be collectively scaled up or down based on their emission share and spatial distribution in the underlying inventory.

- How is the TROPOMI methane background subtracted? Background subtraction tends to be a major source of error in regional emission estimation.

The background consists of a constant value, a temporal linear increase, a seasonal cycle, a daily signal, and a horizonal signal. This encompasses the consideration of both temporal and spatial variations in the background removal. The description of background removal has been added in Section 2.4.

"It is of importance to separate the increase of the atmospheric $CH_4$ concentration due to local emissions from the accumulated atmospheric $CH_4$ background concentration (the $CH_4$ atmospheric lifetime is in the order of 12 years). A Jacobian matrix is introduced to reconstruct the background according to a few background model coefficients, i.e., a constant $CH_4$ value and superimposed disturbances: a temporal linear increase, a seasonal cycle determined by the amplitude and phase of the three frequencies 1/year, 2/year and 3/year, a daily signal (same value for all data measured during a single day), and a horizonal

gradient (same value for any time but dependent on the horizontal location) (Tu et al., 2022a). In the following discussion, the satellite enhancements refer to the residual signal as deduced from TROPOMI CH$_4$ observations after subtracting the modelled background (Figure 4 lower panel)."

The referee is right that the background subtraction can be a major source of error in emission estimation. To further study the impact of background subtraction on emission estimates, the 10$^{th}$ lower percentile of overall satellite observations each day is considered as an alternative choice for setting the background value for the study area on that day. The enhancements (TROPOMI XCH$_4$ - background) and the wind-assigned anomaly computed from the enhancements based on the 10$^{th}$ percentile are compared to those values based on the spatial and temporal variation (default calculation in this study), as presented below. The gridded enhancements computed from the 10$^{th}$ percentile show higher values (21.5 ppb in average) than those based on the spatial and temporal variation method. When comparing the resulting wind-assigned anomalies, an excellent correlation with a R$^2$ value of 0.98 and a slope near unity is found. The wind-assigned anomalies approach, which derives emissions from differences of XCH$_4$ observations associated with opposite wind orientations, helps to effectively mitigate systematic errors associated with background subtraction.

[Figure]

*Figure 1: XCH$_4$ enhancements (TROPOMI - background) and its corresponding wind-assigned anomaly using different background removal methods. The grey line corresponds to the 1:1 line.*

- L. 231: What "calculated average" value is being reported here? It's unclear what the ppb values refer to.

The "calculated average" refers to the observed enhancement (i.e., TROPOMI XCH$_4$ – background). This information has been detailed in the text. The a-priori inventory information, including the location and emission rates of the sources, has a small impact on the estimation of the background, as illustrated in the left figure below. The difference ($1.12 \pm 2.93$ ppb, $R^2 = 0.8562$) arising from the use of different inventories as the a priori, is effectively mitigated ($R^2 = 0.9962$) when comparing the wind-assigned anomalies, as illustrated in the right figure below. It is because the systematic errors in background removal is compensated by computing the differences of enhancements under different wind field segmentations. These figures are presented in Figure A-14 in the updated manuscript.

[Figure]

- The uncertainty analysis varying the plume model, wind product, and inventory is a good start, but not sufficient. What is the sensitivity to background subtraction scheme? What about wind speed levels (e.g., using the 10 m wind rather than 100 m)?

To assess the impact of background removal sensitivity, the study now employs the 10th lower percentile of overall satellite observations as alternative choice for the daily background in the study area, deviating from the approach outlined in Section 2.3, which separately considers spatial and temporal variations. The substitution of the background removal method results in a 7% increase in estimated emission rates in Changzhi, a 6% increase in Jincheng and a 9% increase in Yangquan.

Lower estimates are observed in three regions when employing the wind at 10 m. Specifically, there is a 12% decrease in estimation strength in Changzhi, 11% in Jincheng and 4% in Yangquan. These differences can be attributed to reduced wind speed at lower level, resulting in measured lower wind speed of 15% in Changzhi, 17% in Jincheng and 10% in Yangquan.

Both of these sensitivities are thoroughly addressed in the Uncertainty analysis (Section 3)

- How are the current error values calculated? Do they represent 1-sigma errors? Reporting the uncertainty as the range of estimates from a broader estimation ensemble might be clearer.

The current error values derived from the background removal and the satellite noise. To calculate the uncertainty of the background signal, we first compute the difference between the satellite observations ($\mathbf{y}$) and the modeled background ($\mathbf{K}^{*}_{\mathrm{BG}}\hat{\mathbf{x}}_{\mathrm{BG}}$), and then determine the mean square value ($\mathbf{S}_{\mathrm{y,BG}}$) from its elements representing observations unaffected by the plume. The uncertainty of the background model coefficients can be calculated as $\mathbf{S}_{\hat{\mathbf{x}}_{\mathrm{BG}}} = \mathbf{G}_{\mathrm{BG}}\mathbf{S}_{\mathrm{y,BG}}\mathbf{G}^{T}_{\mathrm{BG}}$. The $\mathbf{G}_{\mathrm{BG}}$ is the gain matrix.

The observed wind-assigned anomaly $\Delta\mathbf{y}_{\mathrm{plume}}$ is a column vector, obtained as the product of satellite signals $\mathbf{y}_{\mathrm{plume}}$ and the operate $\mathbf{D}$, which represents the binning, the averaging, the wind-assigned Δ-maps calculations and the data number filtering. The uncertainty covariance can be written as:

$$\Delta\mathbf{S}_{\mathrm{y,plume}} = \mathbf{D}\mathbf{S}_{\mathrm{y,plume}}\mathbf{D}^{T}$$

A Jacobian $\Delta k$ represents the wind-assigned anomaly model, aiding in generation of the wind-assigned anomaly $\Delta y_{\text{plume}}$, i.e., $\Delta y_{\text{plume}} = \Delta k x$. Here, the coefficient $x$ represents the scaling factors for adjusting the a priori emission rates to achieve the best agreement with the observed plume. Thus, a row vector can be derived as:

$$g^T = (\Delta k^T \Delta S^{-1}_{y,\text{plume}} \Delta k)^{-1} \Delta k^T \Delta S^{-1}_{y,\text{plume}}$$

The background uncertainty ($\epsilon_{BG}$) and the noise in the satellite data ($\epsilon_n$) can be estimated as:

$$\epsilon_{BG} = \sqrt{g^T D K_{BG} S_{\hat{x}_{BG}} K^T_{BG} D^T g}$$

$$\epsilon_n = \sqrt{g^T D S_{y,n} D^T g}$$

The uncertainties in the updated manuscript include additional errors introduced by the dispersion model and its input data through the error propagation. This information has been updated in the conclusion section.

- L. 246-250: These sentences are contradictory. Should the first sentence only refer to the "bottom-up" inventory (which, again, does not seem to me to be a "bottom-up" inventory – rather a measurement-based inventory).

Thanks to the referee. The sentences have been modified.

"The estimates obtained derived through the wind-assigned anomaly method demonstrate comparability with the IPCC Tier 2 bottom-up inventory (Qin et al., 2023). Compared to the estimates, the inventory shows relative differences of 31%, -7%, and -12% in Changzhi, Jincheng, and Yangquan, respectively."

**Typos**

- 14: "process" → "progress" ?

corrected.

- 27: "emission" → "emissions"

corrected.

- 51: "emissions" → "emission estimates"

corrected.

- 53: "from satellite" → "from satellites"

corrected.

- 70: "unprecedented high spatial resolution…" → "unprecedented combination of high spatial resolution…"

corrected.

- 6: "bottum" → "bottom"

The typo in the legend of fig. 6 has been corrected.

- 243: "achived" → "achieved" ?

corrected.

- 252: "boarded" → "bordered" ?

corrected.

**References**

Chen, Z., Jacob, D. J., Gautam, R., Omara, M., Stavins, R. N., Stowe, R. C., Nesser, H., Sulprizio, M. P., Lorente, A., Varon, D. J., Lu, X., Shen, L., Qu, Z., Pendergrass, D. C., and Hancock, S.: Satellite quantification of methane emissions and oil–gas methane intensities from individual countries in the Middle East and North Africa: implications for climate action, Atmos. Chem. Phys., 23, 5945–5967, https://doi.org/10.5194/acp-23-5945-2023, 2023.

Cusworth, D. H., Thorpe, A. K., Ayasse, A. K., Stepp, D., Heckler, J., Asner, G. P., Miller, C. E., Chapman, J. W., Eastwood, M. L., Green, R. O., Hmiel, B., Lyon, D., and Duren, R. M.: Strong methane point sources contribute a disproportionate fraction of total emissions across multiple basins in the U.S., Earth ArXiv, https://doi.org/10.31223/X53P88, 2022.

Shen, L., Gautam, R., Omara, M., Zavala-Araiza, D., Maasakkers, J. D., Scarpelli, T. R., Lorente, A., Lyon, D., Sheng, J., Varon, D. J., Nesser, H., Qu, Z., Lu, X., Sulprizio, M. P., Hamburg, S. P., and Jacob, D. J.: Satellite quantification of oil and natural gas methane emissions in the US and Canada including contributions from individual basins, Atmos. Chem. Phys., 22, 11203–11215, https://doi.org/10.5194/acp-22-11203-2022, 2022.

Shen, L., Jacob, D.J., Gautam, R. et al. National quantifications of methane emissions from fuel exploitation using high resolution inversions of satellite observations. Nat Commun 14, 4948 (2023). https://doi.org/10.1038/s41467-023-40671-6